



# Constraining remote oxidation capacity with ATom observations

Katherine R. Travis[1a], Colette L. Heald[1,2], Hannah M. Allen[3], Eric C. Apel[4], Stephen R. Arnold[5], Donald R. Blake[6], William H. Brune[7], Xin Chen[8], Róisín Commane[9], John D. Crounse[10], Bruce C. Daube[11], Glenn S. Diskin[12], James W. Elkins[13], Mathew J. Evans[14,15], Samuel R. Hall[4], Eric J. Hintsa[13,16], Rebecca S. Hornbrook[4], Prasad S. Kasibhatla[17], Michelle J. Kim[10,18], Gan Luo[19], Kathryn McKain[20], Dylan B. Millet[8], Fred L. Moore[13,16], Jeffrey Peischl[16,20], Thomas B. Ryerson[20], Tomás Sherwen[14,15], Alexander B. Thames[7], Kirk Ullmann[4], Xuan Wang[11,21], Paul O. Wennberg[3,18], Glenn M. Wolfe[22], Fangqun Yu[19]

[1]Department of Civil and Environmental Engineering, Massachusetts Institute of Technology, Cambridge, MA, USA
[2]Department of Earth, Atmospheric and Planetary Sciences, Massachusetts Institute of Technology, Cambridge, MA, USA
[3]Division of Chemistry and Chemical Engineering, California Institute of Technology, Pasadena, CA
[4]Atmospheric Chemistry Observations & Modeling Laboratory, National Center for Atmospheric Research, Boulder, Colorado, USA
[5]Institute for Climate and Atmospheric Science, School of Earth and Environment, University of Leeds, Leeds, UK
[6]Department of Chemistry, University of California Irvine, Irvine, CA, USA
[7]Department of Meteorology, Pennsylvania State University, University Park, PA, USA
[8]University of Minnesota, Department of Soil, Water and Climate, St. Paul, Minnesota, USA
[9]Dept. of Earth & Environmental Sciences of Lamont-Doherty Earth Observatory and Columbia University, Palisades, NY
[10]Division of Geological and Planetary Sciences, California Institute of Technology, Pasadena, CA, USA
[11]Harvard John A. Paulson School of Engineering and Applied Sciences, Harvard University, Cambridge, MA, USA
[12]NASA Langley Research Center, Hampton, Virginia, USA
[13]Global Monitoring Division, NOAA Earth System Research Laboratory, Boulder, CO, USA
[14]Wolfson Atmospheric Chemistry Laboratories (WACL), Department of Chemistry, University of York, York, UK
[15]National Centre for Atmospheric Science (NCAS), University of York, York, UK
[16]Cooperative Institute for Research in Environmental Science, University of Colorado, USA
[17]Nicholas School of the Environment, Duke University, Durham, NC, USA
[18]Division of Engineering and Applied Science, California Institute of Technology, Pasadena, CA, USA
[19]Atmospheric Sciences Research Center, University of Albany, Albany, New York, USA
[20]Chemical Sciences Division, NOAA Earth System Research Laboratory, Boulder, CO, USA
[21]School of Energy and Environment, City University of Hong Kong, Hong Kong, China
[22]Atmospheric Chemistry and Dynamics Laboratory, NASA Goddard Space Flight Center, Greenbelt, MD, USA
[a]Now at NASA Langley Research Center, Hampton, Virginia, USA

Correspondence to: K. R. Travis (katherine.travis@nasa.gov) and C. L. Heald (heald@mit.edu)

**Abstract.** The global oxidation capacity, defined as the tropospheric mean concentration of the hydroxyl radical (OH), controls the lifetime of reactive trace gases in the atmosphere such as methane and carbon monoxide (CO). Models tend to underestimate the methane lifetime and CO concentrations throughout the troposphere, which is consistent with excessive OH. Approximately half the oxidation of methane and non-methane volatile organic compounds (VOCs) is thought to occur over the oceans where oxidant chemistry has received little validation due to a lack of observational constraints. We use observations from the first two deployments of the NASA ATom aircraft campaign during July-August 2016 and January-February 2017 to evaluate the oxidation capacity over the remote oceans and its representation in the GEOS-Chem chemical transport model. The model successfully simulates the magnitude and vertical profile of remote OH within the measurement uncertainties.



Comparisons against the drivers of OH production (water vapor, ozone, and $NO_y$ concentrations, ozone photolysis frequencies) also show minimal bias with the exception of wintertime $NO_y$, for which a model overestimate may indicate insufficient wet scavenging and/or missing loss on seasalt aerosol but large uncertainties remain that require further studies of $NO_y$ partitioning and removal in the troposphere. During the ATom-1 deployment, OH reactivity (OHR) below 3 km is significantly enhanced,

and this is not captured by the sum of its measured components ($cOHR_{obs}$) or by the model ($cOHR_{mod}$). This enhancement could suggest missing reactive VOCs but cannot be explained by new estimates of ocean VOC sources and additional modeled reactivity in this region would be difficult to reconcile with the full suite of ATom measurement constraints. The model generally reproduces the magnitude and seasonality of $cOHR_{obs}$ but underestimates the contribution of oxygenated VOC, mainly acetaldehyde, which is severely underestimated throughout the troposphere despite its calculated lifetime of less than

a day. Missing model acetaldehyde in previous studies was attributed to measurement uncertainties that have been largely resolved. Observations of peroxyacetic acid (PAA) provide new support for remote levels of acetaldehyde. The underestimate in modeled acetaldehyde and PAA is present throughout the year in both hemispheres and peaks during Northern Hemisphere summer. The addition of ocean VOC sources in the model increases annual surface $cOHR_{mod}$ by 10 % and improves model-measurement agreement for acetaldehyde particularly in winter but cannot resolve the model summertime bias. Doing so would

require a 100 Tg yr$^{-1}$ source of a long-lived unknown precursor throughout the year with significant additional emissions in the Northern Hemisphere summer. Improving the model bias for remote acetaldehyde and PAA is unlikely to fully resolve previously reported model global biases in OH and methane lifetime, suggesting that future work should examine the sources and sinks of OH over land.

## 1 Introduction

The hydroxyl radical (OH) is the main oxidant responsible for removing trace gases from the atmosphere and its concentration defines the tropospheric oxidation capacity. OH is primarily produced by the photolysis of ozone ($O_3$) in the presence of water vapor. The lifetimes of key atmospheric trace gases are governed by how quickly they are removed by reaction with OH. Oxidation of volatile organic compounds (VOCs) by OH is a key process for formation of both tropospheric ozone and fine particulate matter which are detrimental to human health and vegetation and impact climate. The oxidation of VOCs, in

addition to carbon monoxide (CO) and methane, provides the main sink of OH in the troposphere. Over half of the production of CO results from the oxidation of methane and other VOCs by OH (Duncan et al., 2007; Safieddine et al., 2017) resulting in a tight coupling of these compounds.

Models tend to overestimate global mean tropospheric OH and the ratio of Northern to Southern Hemisphere mean OH (Naik et al., 2013; Patra et al., 2014). These biases may be linked to the persistent CO underestimate in models (Shindell et al., 2006)

as prescribing OH from observations improves the CO simulation (Müller et al., 2018). However, recent efforts to constrain models with observations of ozone and water vapor could not completely resolve excessive model OH (Strode et al., 2015).



Estimates of OH across models vary due to additional factors, including differing chemical mechanisms and methods for calculating ozone photolysis (Nicely et al., 2017) which are more difficult to isolate. Improving the performance of model chemical mechanisms has largely focused over regions of strong biogenic and anthropogenic activity (e.g. Marvin et al., 2017) but over half of the oxidation of methane occurs over the ocean where models have received little evaluation due to a lack of

observational constraints.

The advent of airborne measurements of OH reactivity (OHR) provides a method to evaluate the total sink of OH across a range of altitudes and a variety of locations and chemical environments (Mao et al., 2009). Previous work compared surface observations of OHR at a single site to the sum of individually calculated OHR components from measurements (Di Carlo, 2004; Yoshino et al., 2006; Sinha et al., 2008, 2010; Mao et al., 2010; Dolgorouky et al., 2012; Hansen et al., 2014; Nakashima

et al., 2014; Nölscher et al., 2012, 2016; Ramasamy et al., 2016; Zannoni et al., 2016, 2017) or from simple models (Ren et al., 2006; Lee et al., 2009; Lou et al., 2010; Mogensen et al., 2011; Mao et al., 2012; Edwards et al., 2013; Kaiser et al., 2016; Whalley et al., 2016). Ferracci et al. (2018) explored the impact of missing OHR estimated from surface observations on modeled global OH levels. Safieddine et al. (2017) and Lelieveld et al. (2016) presented the first global model simulations of OHR but with minimal and qualitative observational evaluation. No study has quantitatively compared simulated and observed

remote OHR in a global model in an effort to constrain the OH sink.

The ATom campaign (Wofsy et al., 2018) provides an unprecedented opportunity to evaluate OH in the remote atmosphere with a detailed suite of chemical observations. We use the GEOS-Chem chemical transport model (CTM) to simulate the first two deployments (ATom-1: July-August 2016, ATom-2: January-February 2017) with the goal of reducing the uncertainty in simulating remote tropospheric OH. We specifically focus on model validation with measurements of OHR, a relatively new

constraint available for assessing total atmospheric oxidation. To our knowledge, this is the first quantitative use of this measurement to evaluate a CTM.

## 2 Description of Model and Observations

### 2.1 The GEOS-Chem model

We use the GEOS-Chem global 3-D CTM in v12.3.0 (doi:10.5281/zenodo.2620535) driven by assimilated meteorological

data from the Goddard Earth Observing System Modern-Era Retrospective analysis for Research and Applications, Version 2 (MERRA-2; Gelaro et al., 2017). The native MERRA-2 model has a horizontal resolution of 0.5° x 0.625° and 72 vertical levels which we degrade to 2° x 2.5° and 47 vertical levels for use in GEOS-Chem. We use timesteps of 20 and 10 min for chemistry and transport, respectively, as recommended by Philip et al. (2016). GEOS-Chem includes detailed treatment of $HO_x$-$NO_x$-VOC-halogen-aerosol chemistry with recent improvements for isoprene (Chan Miller et al., 2017; Fisher et al.,

2016; Marais et al., 2016; Travis et al., 2016), peroxyacetyl nitrate (PAN) (Fischer et al., 2014) and halogen chemistry



(Sherwen et al. 2016). Organic aerosol is parameterized using fixed yields from isoprene, monoterpenes, biomass burning, and anthropogenic fuel combustion (Pai et al., 2019). Aerosol uptake of $HO_2$ is parameterized with a reactive uptake coefficient ($\gamma$) of 0.2 (Jacob, 2000) to produce $H_2O$ (Mao et al., 2013). Methane concentrations are calculated using prescribed surface concentrations derived from monthly observations from the NOAA GMD flask network. We simulate the 2016-2017 period

with an 18-month spin-up period.

Global fire emissions for 2016 and 2017, at 3-hourly resolution (Mu et al., 2011), are from the Global Fire Emissions Database (GFED4s; van der Werf et al., 2017). The GFED4s burned area (Giglio et al., 2013) includes a parameterization of small fires (Randerson et al., 2012). Biogenic VOC emissions are from MEGANv2.1 (Guenther et al., 2012; Hu et al., 2015). Global anthropogenic emissions are from the Community Emissions Data System (CEDS) inventory (Hoesly et al., 2018), overwritten

by ethanol from the POET inventory (Olivier et al., 2003; Granier et al., 2005), ethane from Tzompa-Sosa et al. (2017), and regional    inventories    for    the    United    States    (NEI11v1,    Travis    et    al.,    2016),    Canada    (CAC, https://www.canada.ca/en/services/environment/pollution-waste-management/national-pollutant-release-inventory.html), Mexico (BRAVO, Kuhns et al., 2003), Europe (EMEP, http://www.emep.int/index.html), Asia (MIX, Li et al., 2017), and Africa (DICE, Marais and Wiedinmyer, 2016). Lightning emissions are constrained with satellite data according to Murray et

al. (2012) with a revised global flash rate of 280 mol NO flash$^{-1}$ (Marais et al., 2018) for a source strength of 6.0 Tg N yr$^{-1}$. We emit acetaldehyde (Millet et al., 2010), acetone (Fischer et al., 2012), and dimethyl sulfide (Breider et al., 2017) from air-sea exchange of VOCs produced from biogenic activity in the oceans. All emissions are processed using the Harvard Emissions Component (HEMCO, Keller et al., 2014). Table 1 provides the 2016 emission budget for CO and $NO_x$.

We expand the standard simulation (which includes only background methanol concentrations), to include methanol emissions

and chemistry, as well as emissions and chemistry of unsaturated $C_2$ compounds. Air-sea exchange of methanol is specified using the methodology of Millet et al. (2008) with a constant seawater concentration of 142 nM. Terrestrial biogenic methanol emissions are from MEGANv2.1 and anthropogenic and biomass burning emissions are from the inventories described above. We likewise include biomass burning and anthropogenic emissions of ethyne ($C_2H_2$) and ethene ($C_2H_4$) along with terrestrial biogenic emissions of $C_2H_4$ as above. Oxidation of $C_2H_2$ by OH proceeds according to the Master Chemical Mechanism

(MCM) v3.3.1 (Jenkin et al., 1997; Saunders et al., 2003), via: http://mcm.leeds.ac.uk/MCM. Simplified $C_2H_4$ chemistry is included based on Lamarque et al. (2012) with an updated OH rate constant from the MCM v3.3.1. Table S4 shows the reactions and species included for unsaturated $C_2$ compounds. The model does not consider the OH reactivity of a subset of organic acids (RCOOH) from VOC oxidation. We evaluate the impact of excluding this species, which is minor, in Table S5 and Fig. S8.

The GEOS-Chem mean simulated tropospheric OH for 2016 is 11.9 x 10$^5$ molecules cm$^{-3}$ and the corresponding methane lifetime ($\tau_{CH_4}$) is 9.0 years. This result is comparable to the multi-model mean OH of 11.1 $x$ 10$^5$ molecules cm$^{-3}$ and $\tau_{CH_4}$ of





9.7 years from Naik et al. (2013). The best observationally-derived estimate of $\tau_{CH_4}$ is $11.2 \pm 1.3$ years (Prather et al., 2012), suggesting a model bias here of 20 %. The ratio of tropospheric mean OH in the Northern to Southern Hemisphere is 1.12, which exceeds observationally-derived ratios of 0.85-0.97 (Montzka et al., 2000; Patra et al., 2014) but is improved over previous model estimates ranging from 1.13-1.42 (Naik et al., 2013).

## 2.2 Calculated OH reactivity

The atmosphere contains thousands of reactive organic compounds (Goldstein and Galbally, 2007). Transforming the concentrations of these compounds (as well as those for inorganics that react with OH) to calculated OH reactivity (cOHR) ranks them in order of their importance as OH sinks. The cOHR from a model (cOHR$_{mod}$) can then be compared to cOHR from a suite of measurements (cOHR$_{obs}$), where cOHR is defined by Eq. (1). Recent work from Chen et al. (2019) used this framework to compare the reactivity of a suite of VOCs from a model to observations and found that biogenic species dominate emitted VOC reactivity over North America.

$$cOHR(s^{-1}) = k_{OH,CH_4}[CH_4] + k_{OH,CO}[CO] + k_{OH,NO_2}[NO_2] + \sum k_{OH,VOC}[VOC] + \cdots, \tag{1}$$

Figure 1a shows the simulated annual surface cOHR$_{mod}$ for the year 2016 based on the simulated constituents listed in Table S1. Three-quarters of cOHR$_{mod}$ resides below 3 km (Fig. 1b). The average annual surface cOHR$_{mod}$ is 1.8 s$^{-1}$ with approximately 40 % present over the ocean (average of 1.0 s$^{-1}$). Higher cOHR$_{mod}$ occurs in coastal outflow regions and the lowest cOHR$_{mod}$ is present over the Southern Ocean. The maximum cOHR$_{mod}$ (48 s$^{-1}$) appears over northern China due to high concentrations of SO$_2$, NO$_x$, and CO. In the tropics, elevated cOHR$_{mod}$ is mainly due to isoprene, other biogenic species and CO.

## 2.3 ATom observations

The NASA ATom field campaign (Wofsy et al., 2018) sampled the remote troposphere with the DC-8 aircraft from approximately 200 m to 12 km altitude over the Atlantic and Pacific Oceans in four seasons from 2016 to 2018 with the goals of improving the representation of trace gases and short-lived greenhouse-gases in models of atmospheric chemistry and climate. We use data here from the first two deployments (ATom-1 and ATom-2) which sampled winter and summer conditions in each hemisphere. We consider only observations over the ocean (72 % of measurements). Flight tracks for ATom-1 with land-crossings removed are shown in Fig. 2; ATom-2 flight tracks are nearly identical. The model is sampled along the flight tracks and both the model and observations are averaged to the model grid and timestep for all following comparisons.





The aircraft carried an extensive chemical payload including observations of water vapor, methane, CO, OH, $NO_x$, VOCs, photolysis frequencies and OHR. Table 2 describes the observations used in this work.

## 3 Comparison of simulated and measured OH

We compare observed and simulated OH concentrations to evaluate whether differences are consistent with the bias in $\tau_{CH_4}$

discussed in Section 2.1. Figure 3 shows modeled OH sampled along the flight tracks and compared to observed OH (Table 2) for ATom-1 (boreal summer 2016) and ATom-2 (boreal winter 2017) in each hemisphere from the lowest sampled attitude (~200 m) to 10 km. There is no evidence of a systematic overestimate in modeled OH throughout the troposphere. A model OH overestimate is apparent in the lowest two kilometers in the Northern Hemisphere summer, which could indicate excessive OH production or an underestimated sink from ocean VOC emissions. Global models tend to overestimate OH against

constraints from methyl chloroform observations (Shindell et al., 2006; Naik et al., 2013; Nicely et al., 2017) but we find here that tropospheric OH is successfully simulated within observational uncertainty (accuracy of 1.35, $2\sigma$ confidence level).

We calculate the air-mass weighted column average OH ($OH_{col}$) as a metric of the performance of total tropospheric model oxidation. The model $OH_{col}$ concentration is within approximately 20 % of observations during both deployments, with minimal bias (<1 %) during Northern Hemisphere summer when OH is at a maximum. Modeled $OH_{col}$ in the Northern

(Southern) Hemisphere is 4.40 (1.30) molecules $cm^{-3}$ compared against the observations of 4.39 (1.06) molecules $cm^{-3}$ during ATom-1 and 0.94 (2.75) compared against observations of 0.89 (2.46) molecules $cm^{-3}$ during ATom-2. Figure S1 shows the observed frequency distributions of OH which are well-captured by the model. The observed airmass-weighted ratio of Northern to Southern hemispheric OH over the ocean of 4.1 during ATom-1 and 0.36 during ATom-2 indicates a strong seasonality that the model successfully captures (ratios of 3.4 and 0.34), and which is masked by calculations performed on an

annual mean basis (as given in Section 2.1). The seasonality in this ratio reported by Wolfe et al. (2019) for satellite-derived OH during ATom-1 and ATom-2 is more modest because they account for seasonal differences in remote tropospheric air mass between each hemisphere. The successful simulation shown here is consistent with previous success in representing remote OH measurements with simple models during NASA's Pacific Exploratory Mission-Tropics (PEM-Tropics) B campaign in the clean remote Pacific (Tan et al., 2001).

While the model is in good agreement with OH measurements during ATom, the uncertainty in the observations is similar to a recent estimate of the GEOS-Chem model uncertainty for OH (Christian et al., 2018). In addition, the lifetime of OH is short (seconds) and thus atmospheric concentrations are highly variable. As a result, the comparison in Fig. 3 is insufficient to demonstrate model skill in capturing the broader remote oxidation capacity. Good agreement between the model and observations could also result from compensating errors in the OH source and sink. We support the model comparison in Fig.



with an evaluation of the key factors governing OH production and loss measured by ATom and investigate potential missing sources of VOC from the ocean during summertime.

## 4 Constraints on the remote source of OH

Tropospheric OH is primarily produced from the photolysis of ozone in the presence of water vapor (Monks, 2005) and is

enhanced over the ocean by nitrogen oxides ($NO_x$) from lightning and transport from continental sources. Methane, CO, and VOCs provide the main OH sinks (Murray et al., 2014). We compare the model to ATom-1 and ATom-2 observations of the drivers of the tropospheric OH source (water vapor, ozone, ozone photolysis frequency, $NO_x$) to determine possible broader sources of model bias.

Figure 4 compares observations of water vapor mixing ratios to the NASA MERRA-2 reanalysis product used to drive GEOS-

Chem. MERRA-2 was successfully evaluated against recent observations of tropospheric water vapor (Gelaro et al., 2017) and we find similar good model-measurement agreement here for ATom-1 and ATom-2 with no apparent biases throughout the troposphere. Figure 5 compares median ozone photolysis frequencies to evaluate the model treatment of the incoming actinic flux based on MERRA-2 cloud fractions and optical depths. Hall et al. (2018) showed that GEOS-Chem actinic fluxes in both cloudy and clear skies were well simulated during the ATom-1 deployment. The simulations shown in Fig. 5 also show minimal

bias and successfully represents the observed seasonality with summertime values ($\sim 3.4 \times 10^5$ s$^{-1}$) approximately 2.5 times higher than in winter ($\sim 1.3 \times 10^5$ s$^{-1}$).

The GEOS-Chem ozone simulation has been extensively tested against ozonesondes, aircraft, and satellite observations and is largely unbiased (Hu et al., 2017) with the exception of continental surface concentrations (Fiore et al., 2009; Travis et al., 2016). Figure 6 shows that the highest (54-63 ppb) and lowest (14 ppb) tropospheric ozone observed during ATom-1 and

ATom-2 occur during summer in the mid to upper troposphere and marine boundary layer, respectively. Ozone is less variable in wintertime with values between 30-50 ppb. The model generally reproduces the magnitude and shape of the tropospheric ozone profiles as well as the seasonality observed during both deployments. There is no evidence of the systematic Northern Hemisphere ozone bias previously seen in global model evaluations (Young et al., 2013) that was suggested as a cause of excessive OH (Naik et al., 2013). This may be reflected in the improved model hemispheric OH ratio (Section 2.1) seen here

over previous studies. Upper tropospheric ozone is overestimated in winter, but this would not have a large influence on primary OH production at these altitudes.

OH is enhanced in the presence of $NO_x$ ($\equiv NO + NO_2$). We use $NO_y$ here (Fig. 7(a)) as a constraint as observed $NO_2$ was generally near the detection limit in both deployments. We also show NO (Fig. 7(b)) given its key role in secondary OH production. Maximum $NO_y$ occurs in the Northern Hemisphere upper troposphere in summertime due to lightning (Marais et

al., 2018) and the model captures this enhancement. Observations show little variability between summer and winter $NO_y$ in





the lower troposphere. Southern Hemisphere $NO_y$ is underestimated in the lowest few kilometers in both seasons which could be due to missing ocean production of methyl nitrate (Fisher et al., 2018). The largest model discrepancy is an overestimate of approximately 50 % in the Northern Hemisphere wintertime. Observations of NO reflect the structure of $NO_y$, with the exception of in Northern Hemisphere winter. Figure S2 shows that the model $NO_y$ overestimate in this period is driven by a
high bias for nitric acid ($HNO_3$).

Excessive remote $HNO_3$ is a long-standing model deficiency (Bey et al., 2001; Staudt et al., 2003; Brunner et al., 2003, 2005). The model bias identified here is unlikely to result from overestimated continental emissions due to the short lifetime of $NO_y$ against deposition (~3 days in the Northern Hemisphere winter). Models suggest that less than 40 % of emitted $NO_x$ in the U.S. is exported downwind (Dentener et al., 2006; Zhang et al., 2012). The standard model configuration here does not address
the large possible bias in the U.S. anthropogenic $NO_x$ inventory of ~40 % (Travis et al., 2016) or the downward trend in $NO_x$ emissions from Asia of ~30 % since 2011 (Krotkov et al., 2016). Scaling Asia and U.S. $NO_x$ emissions by these percentages improves the model bias in winter by only 15 % below 3 km (Fig. S2). Recent improvements to the simulation of continental wintertime $HNO_3$ (Jaeglé et al., 2018) would similarly be expected to have a marginal effect in our study region.

Kasibhatla et al. (2018) showed that acid displacement of chloride ($Cl^-$) by $HNO_3$ on seasalt aerosol (SSA) could resolve model
overestimates of gas-phase $HNO_3$ in the marine boundary layer using the GEOS-Chem model. A more comprehensive simulation of this process was developed by Wang X. et al. (2019). Figure S2 shows sensitivity tests of this mechanism over the Northern Hemisphere in winter using the mechanism from Wang X. et al. (2019). Their model configuration exhibits a higher larger ozone and smaller $NO_y$ bias when compared to our simulation of the wintertime ATom-2 measurements shown in Figs. 6 and 7; we focus here on relative changes associated with the acid displacement of chloride. Remote $HNO_3$ decreases
by approximately 50 ppt below 1 km which, combined with reduced emissions, would significantly improve the wintertime $NO_y$ bias in this region but the free tropospheric remains. The photolysis of particulate nitrate on coarse-mode SSA ($NIT_s$) resulting from the acid displacement of $Cl^-$ by $HNO_3$ described above has been proposed as a source of $NO_x$ to the marine boundary layer (Ye et al., 2016; Romer et al., 2018) which could counteract the reductions from acid discplament of $Cl^-$ by $HNO_3$. Kasibhatla et al. (2018) implemented photolysis of $NIT_s$ produced from this mechanism to generate NO and HONO in
the marine boundary layer. We add this process to the simulation of Wang X. et al. (2019) at a photolysis frequency of 50 times that of $HNO_3$ (Kasibhatla et al., 2018). As shown in Fig. S2, this mechanism is consistent with observations of NO below 1 km and does not further bias $HNO_3$ but results in increased $NO_y$ due to PAN formation.

The difficulty in resolving the bias in wintertime model $NO_y$ appears to be due to an overestimate in the overall $NO_y$ lifetime as demonstrated by our sensitivities discussed above. Luo et al. (2019) proposed a new treatment of model wet scavenging
using MERRA2 cloud condensation water content and an empirical description of tracer wet removal, as a mechanism to reduce persistent biases in model surface nitrate over the United States (Zhang et al., 2012; Heald et al., 2012). Preliminary tests (Fig. S2) show that revised wet scavenging according to Luo et al. (2019) could fully resolve the remote bias in $HNO_3$





throughout the troposphere. However, this parameterization has received only limited testing over the surface of the continental U.S. and more testing is needed before it can be adopted widely in models. The effect of increased scavenging could have complex effects on global OH due to reduced oxidant loss from heterogeneous chemistry, another area of intensive current research. For example, recent improvements to $N_2O_5$ hydrolysis in cloud water (Holmes et al., 2019) would further increase

tropospheric levels of $HNO_3$ over the current simulation shown here, complicating the results from Luo et al. (2019). Future work should further assess both the validity of the MERRA-2 cloud water product and the robustness of the scavenging mechanism from Luo et al. (2019), combined with improvements to cloud heterogeneous chemistry (Holmes et al., 2019), in the context of all components of $NO_y$ and particulate nitrate throughout the troposphere before any conclusions can be reached about the impact of resolving the model wintertime Northern Hemisphere $NO_y$ bias on global mean OH.

Overall, the main drivers of remote tropospheric OH production are well-simulated in our base-case simulation against the first two ATom deployments with the exception of an $NO_y$ overestimate in the Northern Hemisphere wintertime. Acid displacement of $Cl^-$ by $HNO_3$ on SSA (Kasibhatla et al., 2018; Wang X. et al., 2019) may somewhat improve remote $HNO_3$ but if the resulting nitrate undergoes photolysis (Kasibhatla et al., 2018) the impact on remote $NO_y$ may be negligible. However, both mechanisms require significant further study as tropospheric halogen sources and chemistry and the rate and products of

the photolysis of NITs are highly uncertain. A new parameterization of model wet scavenging (Luo et al., 2019) would greatly improve modeled remote $HNO_3$ and $NO_y$ but also requires substantial further testing against observations of both cloud water and chemical tracers, in combination with recent work on in-cloud hydrolysis of $N_2O_5$ (Holmes et al., 2019).

## 5 Constraints on the remote sink of OH

The primary sinks of tropospheric OH are CO, methane, and VOCs; OHR measurements represent the sum effect of these

species. Previous aircraft measurements of OHR provided evidence of missing reactivity in the remote atmosphere linked to unknown highly reactive VOCs (Mao et al., 2009). We compare OHR during the ATom-1 and ATom-2 deployments to calculated OHR ($cOHR_{obs}$) according to Eq. 1 from the full ATom measurement suite and from the model ($cOHR_{mod}$) sampled along the flight path. Table 2 describes the observations used to calculate cOHR.

Figure 8 shows the comparison of OHR and cOHR from the model and observations. The observed cOHR is typically less

than observed OHR. Along the flight tracks, $cOHR_{obs}$ and $cOHR_{mod}$ show good agreement and high correlation ($r^2$=0.97 for ATom-1 and ATom-2). The model underestimates $cOHR_{obs}$ by up to 15 % in the lowest 3 km; we discuss this difference further below. The measured relationship between OHR and $cOHR_{obs}$ is weaker ($r^2$=0.72 for ATom-1, $r^2$=0.75 for ATom-2). There is an enhancement in OHR near the surface not present in $cOHR_{obs}$ of approximately 0.6 s$^{-1}$ in the Northern Hemisphere and 0.4 s$^{-1}$ in the Southern Hemisphere. This ~30 % discrepancy is not associated with acetonitrile or CO ($r$<0.2) indicating

that biomass burning is not the cause. Acetaldehyde in Northern Hemisphere summer has the strongest relationship with the





missing reactivity ($r$=0.42, *p-value* << 0.01) which suggests a potential role for unmeasured reactive VOCs or their oxidation products.

Ocean emissions of a variety of VOCs may be a source of remote secondary organic aerosol (Gantt et al., 2010; Kim et al., 2017; Mungall et al., 2017). Read et al. (2012) found that missing model oxygenated VOC (OVOC) in the remote marine tropical atmosphere (mainly acetaldehyde) may cause up to an 8% underestimation of the model global methane lifetime due to missing $cOHR_{mod}$. Our base simulation, described in Section 2.1, includes air-sea exchange of acetone, acetaldehyde, dimethyl sulfide and methanol. We test whether additional compounds emitted from the ocean but not generally included in models could increase $cOHR_{mod}$ and improve the observed discrepancy described above. We follow the standard methodology for air-sea exchange described in Millet et al. (2008) to include emission of the species listed in Table 3 using available measurements of each species in seawater, with the addition of isoprene implemented as a direct emission according to Arnold et al. (2009). As shown in Table 3, air-sea exchange represents a net sink of VOC on an annual basis (-68 Tg C yr$^{-1}$) but this is largely driven by ocean uptake of acetone which is not a significant component of cOHR.

Interfacial photochemistry may provide an additional source of abiotic VOC from the ocean. This source is treated separately from air-sea exchange as described above but ocean uptake may still act on these emissions. We model abiotic ocean VOC emissions according to Brüggemann et al. (2018) by applying species-specific emission factors to the monthly ocean photochemical potential derived in their study. We use the emission factor appropriate for the upper bound of this source according to Brüggemann et al. (2017) (Table S2). Table 4 provides a breakdown of these additional VOCs with a total annual emission of 28 Tg C yr$^{-1}$.

Figure 9 shows the annual mean impact of all ocean emissions described in Tables 3 and 4 on $cOHR_{mod}$ by turning off those ocean sources in a one-year simulation. Average annual surface $cOHR_{mod}$ over the ocean increases by 10 % over the base simulation and 15 % over the simulation with no ocean emissions. The largest increases occur in regions of higher biogenic activity along coastlines. The incremental impact of the additional ocean emissions over the base simulation is shown in Fig. S3. Without ocean emissions, global mean OH would be 3 % greater than in the case with comprehensive ocean VOC treatment. Figure 8 shows that along the flight tracks, $cOHR_{mod}$ increases below 3 km by approximately 0.1 s$^{-1}$ in summer and 0.2 s$^{-1}$ in winter which reduces the model bias against $cOHR_{obs}$. The majority of the added species were measured during ATom and would therefore contribute to $cOHR_{obs}$ and cannot explain the gap in OHR.

We evaluate the impact of expanding the oceanic source of reactive VOC to reconcile the discrepancy between $cOHR_{obs}$ and OHR in a similar manner to Mao et al. (2009). Here, we test a source of alkanes as previously suggested (Read et al., 2012), using the model species ALK4 ($\geq C_4$ alkanes) which has a calculated lifetime of less than two days in the Northern Hemisphere summer ($k_{OH}$ = 2.3x10$^{-12}$ cm$^3$ molecules$^{-1}$ s$^{-1}$ at 298 K). Known alkanes have been measured in seawater (Plass-Dülmer et al., 1993) but the implied source is small. Consequently, we use the ALK4 species for testing purposes only. Generating the





missing OHR in this way requires an implausibly large oceanic ALK4 source of 340 Tg C yr$^{-1}$ compared against all other sources of VOC in the model (Tables 3 and 4). A sensitivity test with this source, shown in Fig. 8, largely closes the gap between cOHR$_{mod}$ and OHR but would result in a 20-50 % reduction in OH along the flight tracks, biasing the model OH simulation (Fig. 3) and degrading model NO$_y$ (Fig. 7) due to increased PAN formation. Thames et al. (2019) found that a

partial recycling of OH would be required to maintain consistency with observed OH and HO$_2$ during ATom. If the unknown VOC we suggest includes some OH recycling in its oxidation mechanism, and does not produce PAN, the model bias in OH could be mitigated. We test an additional source of more reactive VOC including OH recycling using isoprene as the new test species by scaling the ALK4 emission source by the reaction rate of isoprene with OH to obtain an emission of approximately 11 Tg C yr$^{-1}$. Figure 7 shows that this source actually has a minimal impact on cOHR$_{mod}$. This is due to the increased reactivity

of CO, acetaldehyde, and other aldehydes in our test with ALK4 that contribute over half of the increase in cOHR$_{mod}$ from both increased production and longer lifetimes from suppressed OH. Reconciling cOHR$_{mod}$ and OHR is difficult using the existing suite of ATom measurement constraints and possible known VOC precursors; further investigation of the accuracy of the OHR measurements in challenging remote conditions may be needed.

We also examine whether the model is able to capture the components of cOHR$_{obs}$ and explore potential additional sources of

missing cOHR$_{mod}$. Figures 10 and 11 show the components of median cOHR in the base simulation below 3 km for each deployment. The composition of cOHR$_{obs}$ is well-represented by the model. CO and methane make up half or greater of both cOHR$_{obs}$ and cOHR$_{mod}$ with a larger contribution in winter when the lifetime of CO is long. During the ATom-1 deployment, cOHR$_{obs}$ is 50 % higher in the Northern Hemisphere (summer) than in the Southern Hemisphere (winter) largely due to the increase in methyl hydroperoxide (MHP) concentrations and the higher reactivity of methane. During the ATom-2 deployment,

cOHR$_{obs}$ is 60% higher in the Northern Hemisphere (winter) than in the Southern Hemisphere (summer) due to the large contribution of CO in Northern Hemisphere wintertime. The model successfully represents the observed seasonality but underestimates cOHR$_{obs}$.

The difference between measured and simulated cOHR is due to difference between measured and simulated concentrations of OVOCs.  These compounds contribute on average 26 % to cOHR$_{obs}$ but only 17 % OF cOHR$_{mod}$. The largest difference in

reactivity is due to the large enhancement in measured acetaldehyde compared with model simulations. Differences between simulated and measured MHP (Fig. S9) are also important and may reflect error in the calculated lifetime (Müller et al. 2016). The differences could however reflect bias in the MHP measurements in the boundary layer (Supplement, Section 9). Due to the measurement uncertainty we do not explore causes of underestimated MHP here. However, inability to reconcile remote acetaldehyde observations with models is a long-standing problem and has been previously observed over the remote ocean

(Singh et al., 2001; Singh et al., 2003; Millet et al., 2010). Singh et al. (2001) proposed that a large, diffuse, and as-yet unknown source of oxygenated compounds such as acetaldehyde must exist in the troposphere. Using observations from Cape Verde, Read et al. (2012) speculated that underestimated model acetaldehyde could be due to alkanes from terrestrial or ocean biogenic





sources. We consider potential missing sources of model acetaldehyde constrained by the ATom measurements over the ocean and assess their potential impact on simulated OH and CO in Section 6.

## 6 Evaluation of the remote source of acetaldehyde

Figure 12 compares the model simulation of acetaldehyde against observations. Average observed concentrations peak in the Northern Hemisphere during ATom-1 with a mixing ratio of 250 ppt despite a lifetime of only several hours in summer. The maximum model underestimate occurs during this period. Observed concentrations are at a minimum during the ATom-2 deployment indicating a strong seasonality in the source. In each deployment, concentrations remain as high as 70-100 ppt as far south as 60°S (Fig. S4) which the model does not reproduce. There is no apparent difference in model bias between observations over the Atlantic or Pacific Ocean (Fig. S5). The model underestimates acetaldehyde on average by more than a factor of two (~50 to 200 ppt) below 3 km and does not capture the observed elevated levels throughout the troposphere, which could support the hypothesis of a missing long-lived precursor suggested by Singh et al. (2001).

In earlier studies, measurement artifacts prevented interpretation of model-measurement disagreements in the remote atmosphere. Previous measurements of acetaldehyde had biases due to difficulties in background subtraction (Apel et al., 2008) with uncertainties as high as 70 ppt (Apel et al., 2003) which hindered analysis of clean conditions. The ATom measurement uncertainty is reduced to 10 ppt/20 % (Table 2) and does not have the biases present in previous campaigns (Wang S. et al., 2019). Previous work disputed whether observed acetaldehyde was compatible with PAN due to the significant role of acetaldehyde as a PAN precursor through production of the peroxyacetyl (PA) radical (Singh et al., 2001; Singh et al., 2003; Millet et al., 2010). Global simulations estimate that acetaldehyde is responsible for approximately 40 % of the production of the PA radical (Fischer et al., 2014), which would be even larger if acetaldehyde is underestimated as suggested above. Reaction of the PA radical with $HO_2$ is more prevalent in remote environments and produces peroxyacetic acid (PAA) preferentially over PAN, making PAA a more useful constraint for the conditions sampled by ATom. Figure 13 shows the model simulation of PAA against observations for each deployment. PAA is underestimated by the model, with the largest model bias during Northern Hemisphere summer, consistent with the model bias in acetaldehyde. Fig. 14 shows the model comparison with PAN, which is generally well simulated.

Wang S. et al. (2019) find using an observationally-constrained box-model that the levels of acetaldehyde observed during ATom are required to explain the observed PAA, although the reaction rate of PAA + OH has an uncertainty of approximately a factor of three. We evaluate the standard GEOS-Chem acetaldehyde budget, described in detail by Millet et al. (2010), against available ATom observations. The 2016 model budget for the base simulation is provided in Table 5. Acetaldehyde is mainly produced from VOC oxidation (ethane, propane, $\geq C_4$ alkanes, $\geq C_3$ alkenes, isoprene, ethanol) and is also directly emitted from



the ocean, terrestrial plant growth, biomass burning, and anthropogenic activities. The parameterization of acetaldehyde ocean emissions is dependent on satellite-based observations of colored dissolved organic matter (CDOM) (Millet et al., 2010).

The model free tropospheric bias suggests that long-lived VOC oxidation must be underestimated due to the short lifetime of acetaldehyde (< 1 day). The longest-lived precursor VOCs in the model are ethane (two months) and propane (two weeks).

Ethane has the highest concentration of any measured non-methane VOC during ATom with an average of 1.5 ppb below 3 km during the Northern Hemisphere winter. The model underestimates ethane and propane by up to 30 % during ATom-1 and 80% during ATom-2 (Figs. S6 and S7, respectively). However, the oxidation of these species is too slow to provide the missing model acetaldehyde and would only marginally increase remote background levels even if it was produced at higher yield at low-$NO_x$ (currently ~50 % for ethane, ~20 % for propane, Millet et al., 2010). The model chemical mechanism for these species

is provided in Table S3. One or more precursors able to resolve the model acetaldehyde bias must therefore be present at higher cumulative concentrations than ethane or propane. Modeled ALK4, parameterized as a butane/pentane mixture, maintains a high acetaldehyde yield at low-$NO_x$ and has a shorter lifetime (~5 days), contributing to a larger perturbation to atmospheric acetaldehyde levels than ethane or propane for a given concentration change. The sensitivity test adding substantial ALK4 emissions from the ocean described in Section 4 would result in only small improvement in the free troposphere but a 50 %

overestimate below 1 km. Furthermore, ALK4 is also too short-lived to substantially perturb the remote atmosphere from a continental source, thus the missing acetaldehyde precursors (from either a marine or terrestrial source) must have a longer lifetime.

As shown in Table 5, primary ocean emissions in the base simulation (22 Tg yr$^{-1}$) are lower than previous work (57 Tg yr$^{-1}$) due to updates to the model parameterization of the water transfer velocity (Johnson, 2010). Additional independent estimates

of the ocean source are also much larger (34-42 Tg yr$^{-1}$, Read et al., 2012; Wang S. et al, 2019). However, an increased primary ocean source would not address the bias in the free troposphere or in winter when biogenic activity from CDOM is zero in the model at high latitudes. Ship-borne measurements generally measure non-zero acetaldehyde seawater concentrations of approximately 5 nM (Read et al., 2012) and a recent trans-Atlantic campaign found that acetaldehyde concentrations from 47°S to 50°N did not always correlate with levels of CDOM (Yang et al., 2014). Therefore, we set a minimum seawater

concentration of 5 nM in the model parameterization regardless of CDOM level. This change adds 2 Tg C yr$^{-1}$ in emissions and increases concentrations over the remote ocean in winter by up to 50 ppt.

Figure 12 shows the combined effect of adding new ocean VOCs in Section 4 and improving the seawater parameterization described above on modeled acetaldehyde (labelled as "Improve Ocean VOCs"). Although the direct ocean source in this work is lower than previous estimates as described above, the secondary source from precursor VOCs is enhanced. Of the additional

marine VOCs described in Section 4, 25 Tg C yr$^{-1}$ produce acetaldehyde as an oxidation product (Tables 3 and 4). This is compared to 12 Tg C yr$^{-1}$ of direct emissions in the base model. These sources substantially increase near-surface simulated acetaldehyde, with the largest improvement during winter (40-70 ppt) when atmospheric lifetimes are longer and the influence



of the ocean can extend aloft. In summer, impacts of 10-60 ppt are limited to the lowest model layer due to higher OH. Recent work over North America suggested that free tropospheric VOC may be underestimated due to errors in model vertical mixing (Chen et al. 2019), but in Northern Hemisphere summer slower mixing would not be expected to compensate for the short lifetime of acetaldehyde in this region (~4 hours). Thus the pervasive model bias in the free troposphere cannot be explained

by an increase in known direct or indirect ocean sources.

Photodegradation of organic aerosol (OA) is another potential source of oxygenated VOCs such as acetaldehyde to the troposphere (Kwan et al., 2006; Epstein et al., 2014; Wong et al., 2015; Wang S. et al., 2019). A previous study suggests that the source of secondary organic aerosol (SOA) would need to be up to four times larger than current estimates given an implied underestimate of the photochemical loss term (Hodzic et al., 2016). We test the potential impact of such a source on

acetaldehyde using the model simulation of OA described in Section 2.1 and increase the overall model production of SOA by a factor of four to maximize the impact of R2 below. We apply a photolysis frequency for OA of $4 \times 10^{-4} J_{NO_2}$ (Hodzic et al., 2015) to the reactions R1 and R2 as an upper limit and describe the formulation of R1 and R2 below.

$$OCPI + h\upsilon = 0.5 \; ALD2 \tag{R1}$$

$$SOAS + h\upsilon = 0.66 \; SOAS + ALD2 \tag{R2}$$

The species OCPI and SOAS represent the majority of modeled OA in the remote atmosphere. OCPI is aged (hydrophilic) organic carbon (12 g C mol$^{-1}$) and SOAS is SOA from all emission categories (150 g mol$^{-1}$). Both are assumed for the purposes of the sensitivity tests here to have an OA/OC ratio of 2.1. In R1, one molecule of carbon (0.5 ALD2) is produced per reaction and in R2, one acetaldehyde molecule (ALD2) is produced per reaction. The resulting impact on modeled acetaldehyde is only appreciable in the Northern Hemisphere winter (Fig. 12) when modeled aerosol amounts are highest and the lifetime of

acetaldehyde is long. Given that this test represents an upper limit, we conclude that organic aerosol photolysis cannot provide a sufficient source of acetaldehyde to reconcile the model with observations.

We consider whether an entirely unknown VOC with moderate lifetime and a high yield of acetaldehyde at low NO$_x$ could resolve the free-tropospheric model bias. We emit such a species with a lifetime of approximately one month against oxidation by OH, emissions of 100 Tg yr$^{-1}$ from either anthropogenic, biomass burning, or ocean sources, and a yield of 1 acetaldehyde

molecule per reaction with OH. We do not test a terrestrial biogenic source here but expect the results would be similar to the biomass-burning case. These simulations result in concentrations of 1-4 ppb of the precursor VOC throughout the troposphere. The effect of the unknown VOC is compatible with the model simulation of OH (unlike the addition of oceanic ALK4 needed to reconcile OHR observations as described in Section 5). Summertime tropospheric OH below 3 km decreases by approximately 6 % against ATom observations over the case with improved ocean emissions, well within measurement and

model uncertainty. The maximum cOHR of this species is small (0.04 s$^{-1}$). The impact on modeled acetaldehyde (Fig. 12) is





generally similar across all three source categories due to the long lifetime of this precursor. As shown in Fig. 12 and 13 the addition of this unknown VOC modestly improves the simulation of acetaldehyde and PAA everywhere but a large residual underestimate in Northern Hemisphere summer remains. The impact on PAN is minor with the exception of Northern Hemisphere winter (Fig. 14), but this is likely driven by the model overestimate in $NO_y$ (Fig. 7).

VOC emissions inventories are known to be incomplete, for example missing emissions from volatile consumer products (McDonald et al., 2018) and biomass burning (Akagi et al. 2011), both of which peak in summer. In the case of fire emissions, half of VOC emissions are unidentified (Akagi et al., 2011) and the average emission factor for this unidentified VOC roughly corresponds to 76 Tg yr$^{-1}$ of unidentified VOC, similar to our sensitivity tests of 100 Tg yr$^{-1}$ described above. However, recent attempts to quantify unidentified VOC from fire (Stockwell et al., 2015; Koss et al., 2018) find that newly identified compounds

tend to be too reactive to impact the remote atmosphere, as needed here, but this work is ongoing and future efforts should investigate potential precursors of acetaldehyde that could be transported to the remote atmosphere. The missing source of precursor VOC must have substantial additional summertime emissions above and beyond the sensitivity tests shown in Fig. 12 to address the Northern Hemisphere summertime bias. The required magnitude of this perturbation will be difficult to reconcile within known measurement and emission uncertainty constraints.

**7 Conclusions**

The rich set of chemical information available from the ATom field campaign provides the most comprehensive dataset ever collected to evaluate models in the remote atmosphere. The sampling strategy of collecting observations throughout the troposphere in multiple seasons is ideally suited for improving our understanding of tropospheric chemistry in a poorly observed region of the atmosphere. We use the first two deployments of the ATom field campaign during July-August 2016

and January-February 2017 to investigate sources of bias in model simulations of OH. Global models such as the GEOS-Chem CTM used here tend to overestimate the loss of methane by OH and underestimate CO which provides the main tropospheric sink of OH. Comparisons of the model with observations from the first two ATom deployments do not show systematic bias in the simulation of OH or the drivers of remote OH production (water vapor, photolysis of ozone, ozone and $NO_y$) with the exception of wintertime $NO_y$ which is biased high by a factor two.

The model overestimate of wintertime $NO_y$ is largely attributable to nitric acid. This bias is not due to an anthropogenic inventory overestimate but may reflect insufficient wet scavenging as well as loss to seasalt aerosol by nitric acid although the former mechanism may be counteracted by photolysis of the resulting nitrate aerosol. The impact of resolving this wintertime $NO_y$ bias on remote OH is uncertain. Future work should improve constraints on these mechanisms, which have all received only preliminary validation, and carefully examine their impact in the context of broader atmospheric chemistry, particularly

$NO_y$ partitioning throughout the troposphere.





We present the first comparison of measured OH reactivity (OHR) from aircraft with a global model to evaluate the tropospheric sink of OH. We calculate OH reactivity (cOHR$_{obs}$) from relevant species observed during ATom and compare this to cOHR from the model (cOHR$_{mod}$). Measured OHR is higher than cOHR$_{obs}$ by approximately 30 % below 3 km. This missing OHR correlates with acetaldehyde during summer indicating a potential source of missing reactive VOC, similar to the findings of Mao et al. (2009) for the NASA INTEX-B field campaign. The addition of a comprehensive set of ocean VOC emissions increases global mean cOHR by 10 % but cannot reproduce the observed OHR enhancement during ATom-1. Adding sufficient alkanes to the model to resolve this bias requires an improbably large ocean VOC source (340 Tg C yr$^{-1}$) and would degrade the model simulation of OH and NO$_y$.

The model successfully simulates the seasonality and hemispheric gradient in cOHR but has a persistent underestimate of up to 15 % in the lowest 3 km, primarily due to an acetaldehyde underestimate. The inability to reproduce observations of remote acetaldehyde was first observed during the PEM-Tropics campaign (Singh et al., 2001; Singh et al., 2003; Millet et al., 2010) but the measurement was uncertain. Improvements in measurement precision and the accompanying measurement of PAA during ATom strengthen the conclusion that there is a large amount of acetaldehyde present in the atmosphere that cannot be explained by current models. We investigate possible underestimates in known sources of acetaldehyde including VOC emissions from anthropogenic, biomass, or oceanic sources or production from the photolysis of organic aerosol. No known source can fully resolve the bias in acetaldehyde throughout the troposphere, and particularly in the Northern Hemisphere summer. We consider the possibility that there is a large, diffuse source of unknown VOC by implementing 100 Tg yr$^{-1}$ of such a compound from ocean, biomass burning, or anthropogenic sources. This hypothetical source modestly reduces the model acetaldehyde bias and is compatible with the simulation of OH and cOHR; however, an additional source is required to resolve the largest bias in the Northern Hemisphere summer. Errors or omissions in the oxidation mechanism of known VOCs may also contribute to this bias. For example, significant uncertainties exist in peroxy radical (RO$_2$) chemistry for large RO$_2$ molecules (Praske et al., 2017), although the flux of carbon through a minor pathway would have to be large, restricting the possible known sources. Further laboratory and field observations are needed to understand which precursors and sources could lead to the sustained production of acetaldehyde observed during ATom and prior campaigns.

This study demonstrates that long-standing model biases in global mean OH are unlikely to be due to errors in simulating tropospheric chemistry over the ocean. This implies that a large bias must be present in OH production or loss over land and future work should focus on evaluating continental OH sources and sinks. Errors in modeled OH were recently investigated by Strode et al. (2015) and when overestimates related to production terms were corrected, model OH remained too high in the Northern Hemisphere, suggesting that future studies should focus on errors in OH loss.



*Author Contributions.* CLH and KRT designed the study. KRT modified the code, performed the simulations and led the analysis. HMA, ECA, DRB, WHB, RC, JDC, BCD, GSD, JWE, SRH, EJH, SRH, MJK, KM, FLM, JP, TBR, ABT, KU, POW, GMW provided ATom measurements used in the analysis. XW provided the model code for the sensitivity runs including acid displacement of chloride on coarse-mode seasalt aerosol. TS, ME and PSK provided the model code for the photolysis of particulate nitrate. GL and FY were responsible for the code for the revised treatment of wet scavenging in the model. DBM and XC provided the methanol seawater concentration and assisted in the ocean budget analysis. SRA provided the biogenic ocean isoprene emissions. KRT and CLH wrote the paper with input from the co-authors.

*Competing interests.* The authors declare that they have no conflict of interest.

*Funding*. CLH and KRT acknowledge from the National Science Foundation (AGS-1564495) and the National Oceanic and Atmospheric Administration (NA18OAR4310110). DBM and XC acknowledge support from the National Aeronautics and Space Administration (NNX14AP89G). TBR was supported by the National Aeronautics and Space Administration (IAT NNH15AB12I) and by funding from the National Oceanic and Atmospheric Administration Climate Program Office and AC4 program. RH and EA were supported by the National Center for Atmospheric Research, which is a major facility sponsored by the National Science Foundation under Cooperative Agreement No. 1852977. GL and FY acknowledge support from the National Aeronautics and Space Administration (NNX17AG35G). Caltech authors acknowledge support from the National Aeronautics and Space Administration (NNX15AG61A).

*Data Availability.* The ATom-1 and ATom-2 data (Wofsy et al., 2018) are available here doi:10.3334/ORNLDAAC/1581.

*Acknowledgements.* We acknowledge helpful conversations and advice from Andrea Molod, Rachel Silvern, Eloïse Marais, Sarah Safieddine, Martin Brüggemann, Christian George and James Crawford. We acknowledge Tom Hanisco for the use of his formaldehyde observations from ATom and Barbara Barletta and Simone Meinardi for their contribution to the UCI WAS measurements.

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





**Table 1**. Annual emissions of CO and $NO_x$ for 2016 used in the GEOS-Chem simulations.

| Emissions category | CO, Tg | Emissions category | $NO_x$, Tg N |
|---|---|---|---|
| Fuel combustion[1] | 590 | Fuel combustion[1] | 32.9 |
| Biomass burning | 311 | Biomass Burning | 6.0 |
| NMVOC Oxidation | 689 | Soil Emissions | 7.8 |
| Methane Oxidation | 938 | Lightning emissions | 6.0 |
| Total | 2528 | Total | 52.7 |
| [1]Anthropogenic fossil fuel and biofuel combustion | | | |

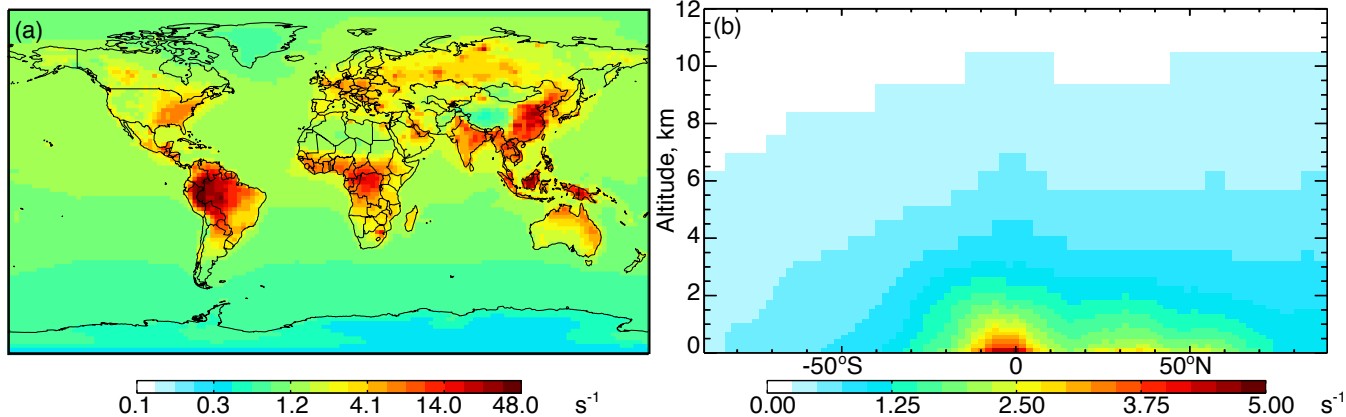

**Figure 1.** Annual mean 2016 a) surface (log scale) and b) zonal mean cOHR calculated from individual model species. The GEOS-Chem species included in the calculation of cOHR are listed in Table S1.

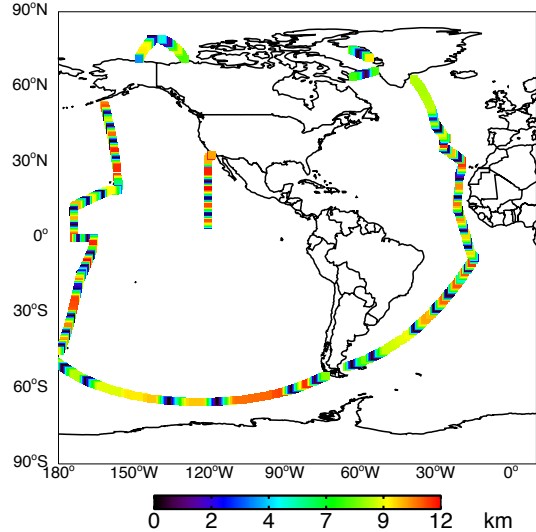

**Figure 2.** ATom-1 ocean-only flight tracks colored by altitude.



**Table 2.** Description of ATom measurements used to evaluate the model simulation.

| Measurement | Instrument | Accuracy | Detection Limit/Precision | Reference |
|---|---|---|---|---|
| OHR | Airborne Tropospheric Hydrogen Oxides Sensor (ATHOS) | 0.8 s$^{-1}$ | $\pm$ 0.3 s$^{-1}$ | Faloona et al., 2004; Mao et al., 2009 |
| Water vapor | Diode laser hygrometer (DLH) | 5% | 0.1% or 50 ppb | Diskin et al., 2002 Podolske et al., 2003 |
| NO$_y$[6] | NOAA Nitrogen oxides and ozone (NO$_y$O$_3$) | | 0.05 ppb[4] | Pollack et al., 2010; Ryerson et al., 1998, 2000 |
| Photolysis frequencies via actinic flux | Charged-coupled device Actinic Flux Spectroradiometers (CAFS) | jO$_3$ 20% jNO$_2$ 12% | jO$_3$ 10$^{-7}$ /s jNO$_2$ 10$^{-6}$ /s | Shetter and Mueller, 1999, Petropavloskikh, 2007, Hofzumahaus et al., 2004 |
| Peroxyacetyl nitrate (PAN) | PAN and Trace Hydrohalocarbon ExpeRiment (PANTHER) | 10% | 2 ppt $\pm$ 10 % | Elkins et al., 2001; Wofsy et al., 2011 |
| **Components of OH reactivity[7]** | | | | |
| CH$_4$ | NOAA Picarro | 0.6 ppb | 0.3 ppb | Karion et al., 2013 AMT |
| CO | Harvard Quantum Cascade Laser System (QCLS) | 3.5 ppb | 0.15 ppb | McManus et al., 2005; Santoni et al., 2014 |
| H$_2$[1] | UAS Chromatograph for Atmospheric Trace Species (UCATS)/PANTHER | | 7.5 ppb[3] | Hintsa et al., 2019 |
| NO, NO$_2$, O$_3$ | NOAA NO$_y$O$_3$ | | 0.006 ppb[4], 0.03 ppb[4], 1.7 ppb[4] | Pollack et al., 2010; Ryerson et al., 1998, 2000 |
| Methyl hydroperoxide, nitric acid, hydrogen peroxide, peroxyacetic acid, peroxynitric acid | Caltech Chemical ionization mass spectrometer (CIMS) | $\pm$ 30 %, $\pm$ 30 %, $\pm$ 30 %, $\pm$ 50 %, $\pm$ 30 % | 25 ppt, 50 ppt, 50 ppt, 30 ppt, 100 ppt | St. Clair et al., 2010; Crounse et al., 2006 |
| Formaldehyde | NASA In Situ Airborne Formaldehyde (ISAF) | 10% | 10 ppt | Cazorla et al., 2015; DiGangi et al., 2011; Hottle et al., 2009 |
| Methanol, acetaldehyde, propane, dimethyl sulfide, ethanol, acetone, methyl ethyl ketone, propanal[5], butanal[5], toluene, methyl vinyl ketone, methacrolein<br><br>i-Butane + n-butane + i-pentane + n-pentane[2] | NCAR Trace Organic Gas Analyzer (TOGA) | 30%, 20%, 30%, 15%, 30%, 20%, 20%, 20%, 30%, 15%, 20%, 20%<br><br>15%, 15%, 15%, 15% | 10 ppt, 10 ppt, 20 ppt, 2 ppt, 30 ppt, 10 ppt, 2 ppt, 20 ppt, 2 ppt, 0.6 ppt, 4 ppt, 2 ppt<br><br>2 ppt, 2 ppt, 4 ppt, 4 ppt | Apel et al., 2015 |
| OH, HO$_2$ | ATHOS | | factor of 1.35 | Faloona et al., 2004 |





| Ethane, benzene | UCI Whole air sampler (WAS) | 5%, 5% | 3 ppt, 3 ppt | Colman et al., 2001; Simpson et al.; 2010 |
| --- | --- | --- | --- | --- |

[1]The GEOS-Chem concentration of $H_2$ is set to a constant value of 0.5 ppm.

[2]Lumped as >$C_4$ alkanes (ALK4) in GEOS-Chem.

[3]Average of reported error for each individual measurement for ATom-1 and ATom-2.

[4]Average of 2-sigma uncertainty for each individual 1 Hz measurement for ATom-1 and ATom-2.

5 [5]Lumped as >$C_3$ aldehydes (RCHO) in GEOS-Chem.

[6]Model $NO_y$ is defined as $NO + NO_2 + HONO + HNO_3 + HNO_4 + 2*N_2O_5 + ClNO_2 + \sum PNs + \sum ANs$.

[7]Included in cOHR are observations of species where at least 20% of the possible available measurements below 3 km are not missing.

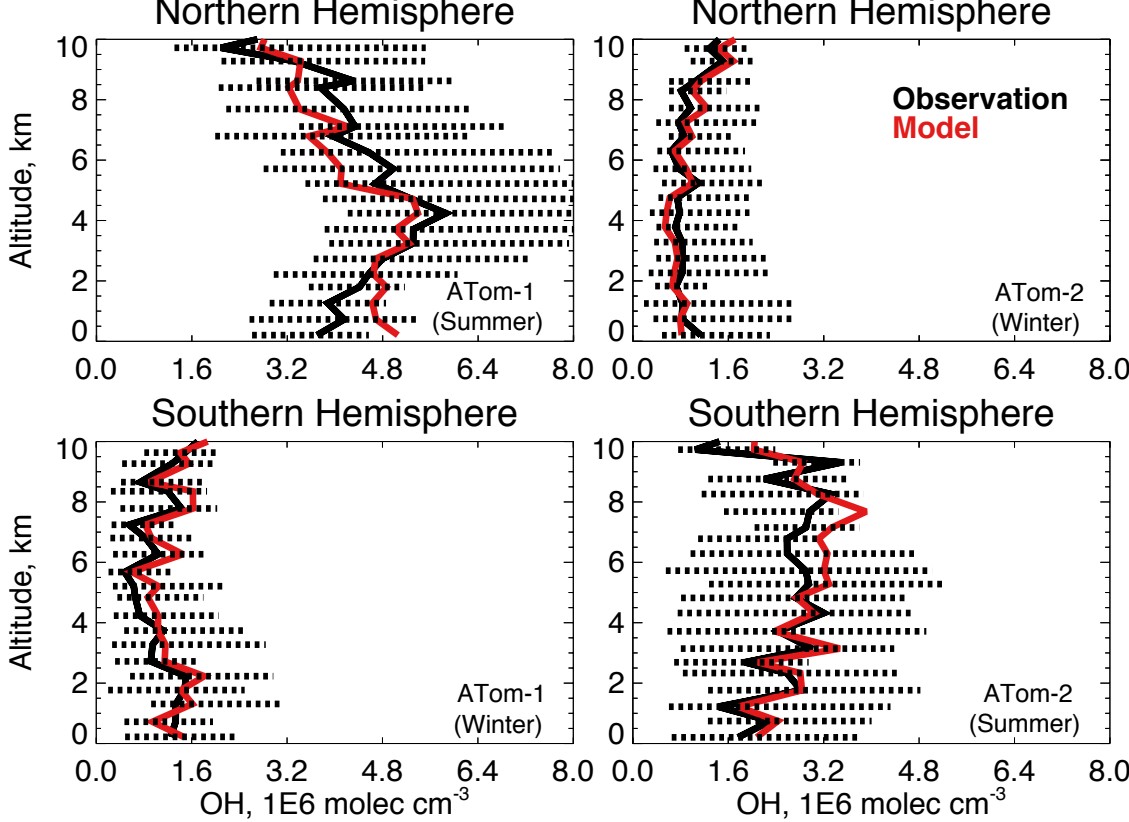

**Figure 3.** Median OH concentrations for the Northern hemisphere (>0°N) and Southern hemisphere (<0°S) from the ATHOS instrument described in Table 2 during ATom-1 (Jul-Aug, 2016) and ATom-2 (Jan-Feb, 2017) compared against the GEOS-





Chem model in 0.5 km altitude bins. The observations have been filtered to remove biomass burning (acetonitrile >200 ppt) and stratospheric ($O_3/CO > 1.25$) influence. The dashed lines show the observed 25th-75th percentiles.

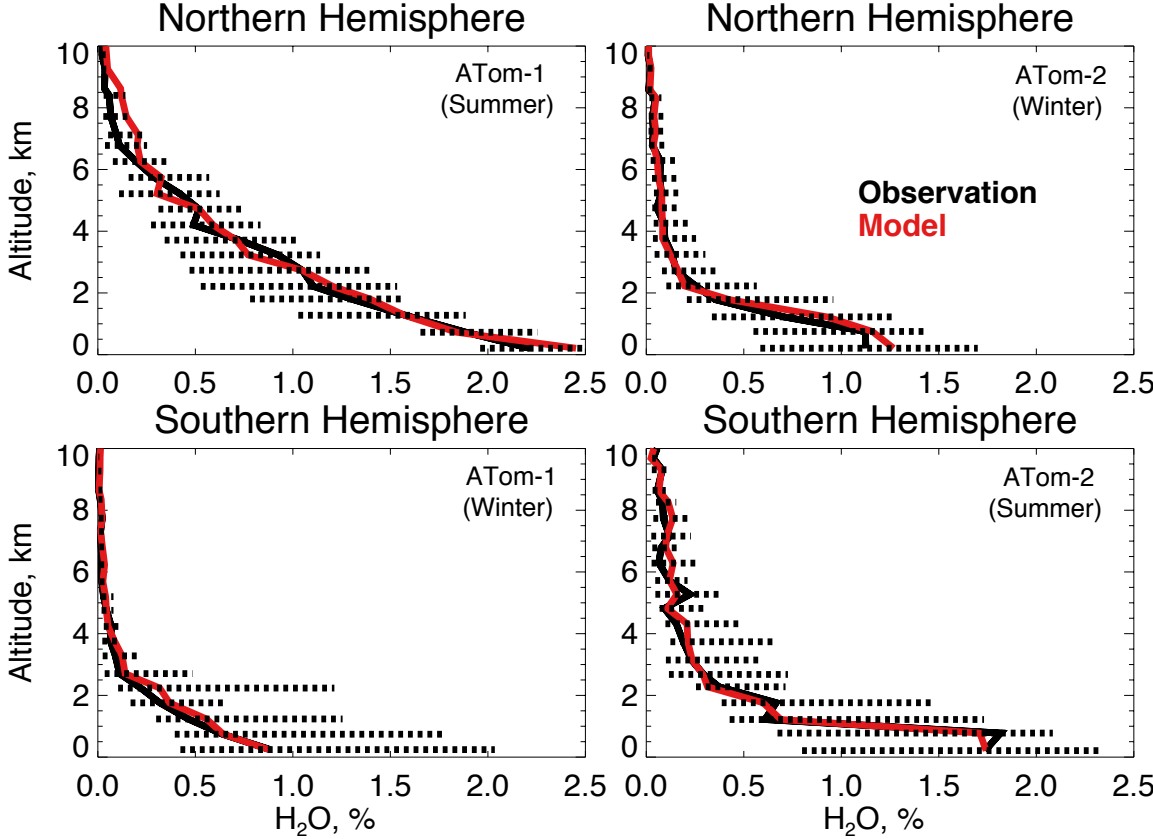

**Figure 4.** The same as Figure 3 for median water vapor concentrations. Water vapor mixing ratio was measured by the DLH instrument as described in Table 2.



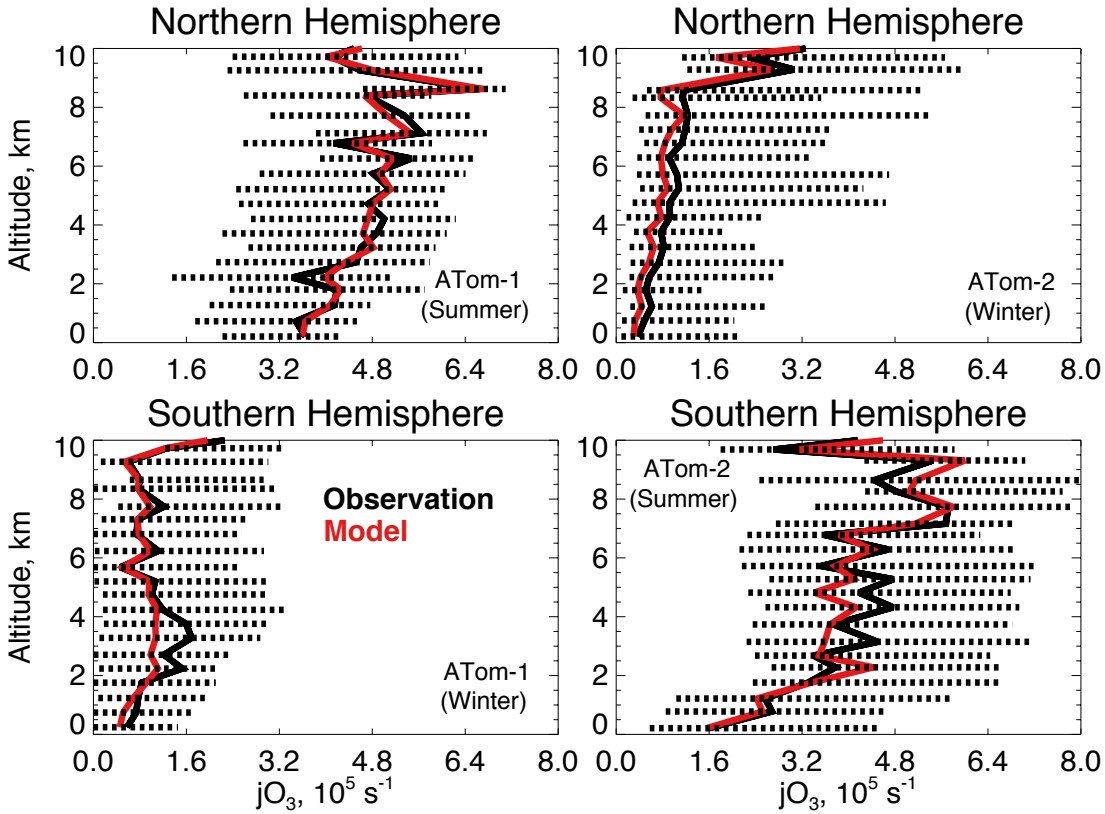

**Figure 5.** The same as Figure 3 for median photolysis frequencies for ozone ($jO_3$). $jO_3$ was determined from actinic flux measured by the CAFS instrument as described in Table 2.



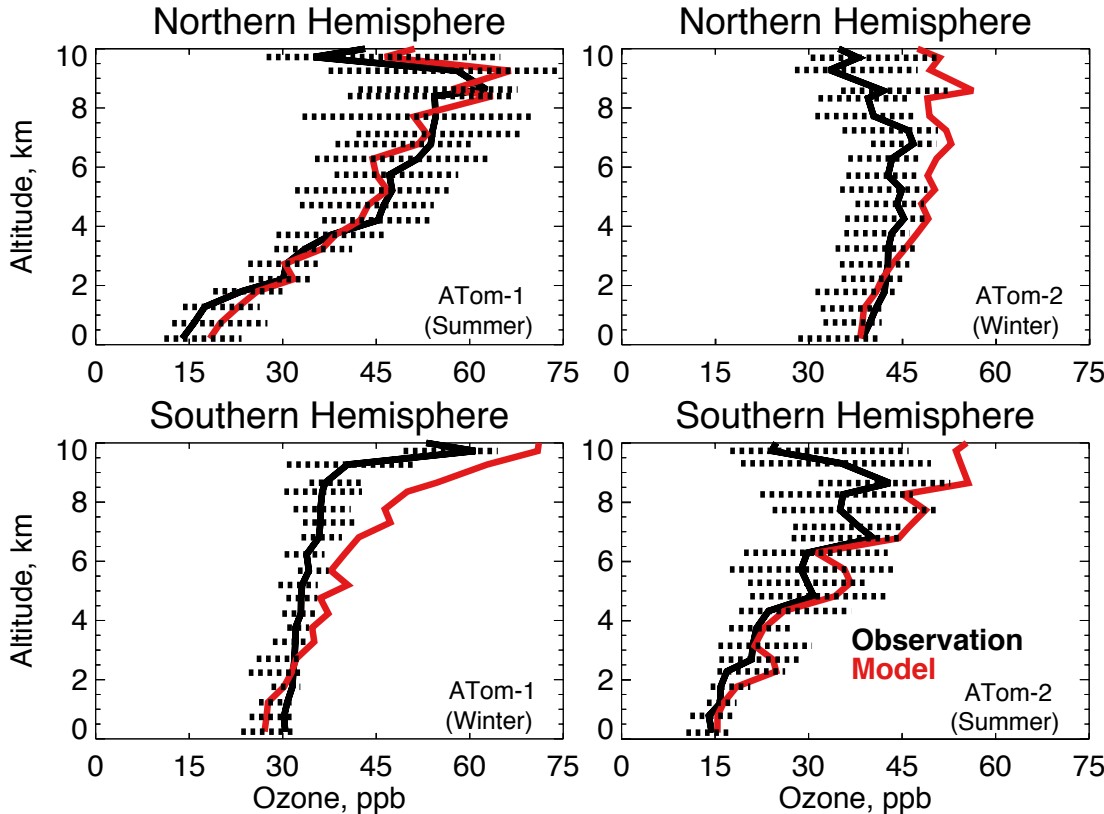

**Figure 6.** The same as Figure 3 for median ozone concentrations. Ozone was measured by the NOAA NO$_y$O$_3$ instrument as described in Table 2.

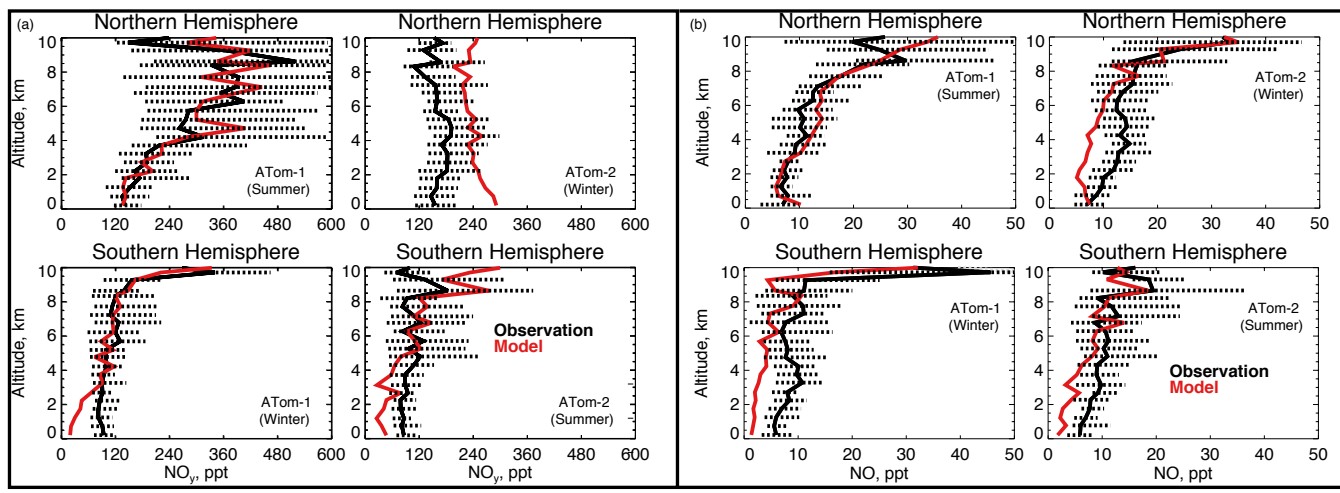

**Figure 7.** The same as Figure 3 for median NO$_y$ (a) and NO (b) concentrations. NO$_y$ and NO were measured by the NOAA NO$_y$O$_3$ instrument as described in Table 2.





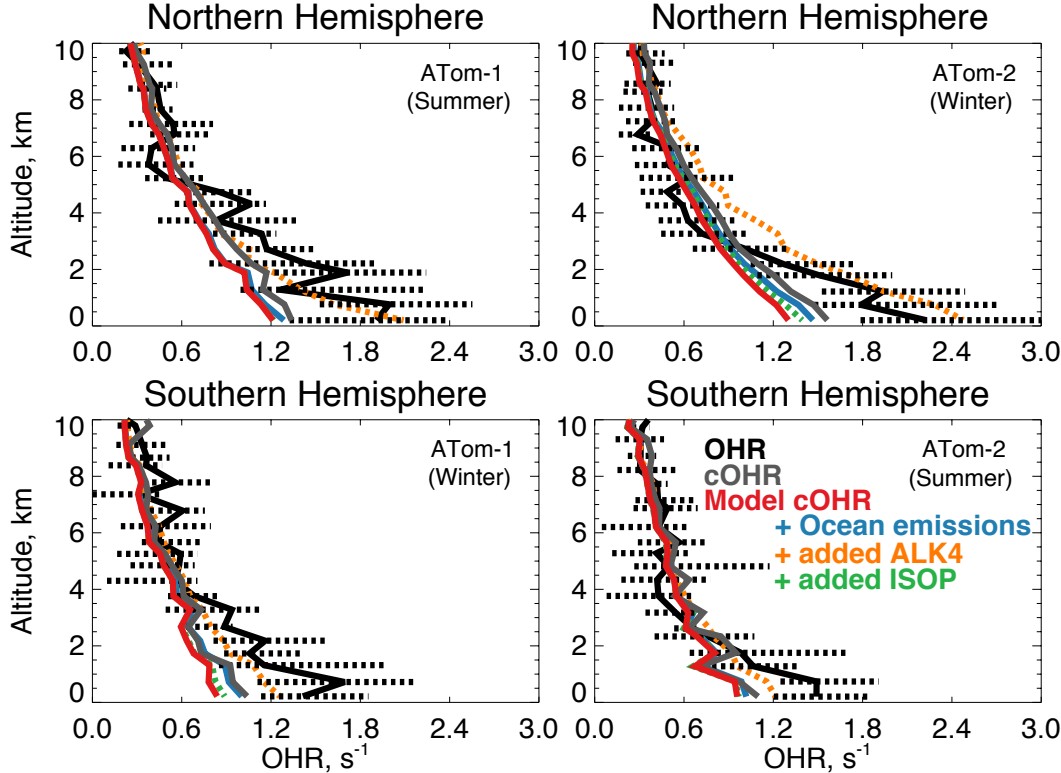

**Figure 8.** The same as Figure 3 for median OHR. OHR was measured by the ATHOS instrument as described in Table 2. The calculation of cOHR in the model and observations includes the species described in Table 2. In order to allow for a point-by-point comparison of cOHR in the model and observations, missing values are filled in the observational components of cOHR using linear interpolation. All calculated reactivity values are determined using the temperature and pressure of the ATHOS instrument inlet which differs from ambient values. The sensitivity tests are described in Section 5.

**Table 3.** Biogenic ocean VOC emissions

| GEOS-Chem species[2] | # lumped species | Produces acetaldehyde? | Annual Net Emissions (Tg C) | Reference for seawater concentration |
|---|---|---|---|---|
| ALD2 | 1 | Yes | 10.25 | Millet et al., 2010 |
| MOH | 1 | No | -1.55 | Pers. comm. D. Millet |
| ACET | 1 | No | -75.70 | Fischer et al., 2012 |
| LIMO | 1 | Yes | 0.04 | Hackenberg et al., 2017 |
| MTPA | 3 | Yes | 0.05 | Hackenberg et al., 2017 |
| MTPO | 2 | Yes | 0.06 | Hackenberg et al., 2017 |
| EOH | 1 | Yes | -5.60 | Beale et al., 2010 |
| C2H6 | 1 | Yes | 0.33 | Plass-Dülmer et al., 1993 |
| C2H4 | 1 | No | 0.75 | Plass-Dülmer C. et al., 1993 |
| PRPE | 2 | Yes | 0.95 | Plass-Dülmer C. et al., 1993 |
| C3H8 | 1 | Yes | 0.16 | Plass-Dülmer et al., 1993 |



| GEOS-Chem species[2] | # lumped species | Produces acetaldehyde? | Annual Net Emissions (Tg C) | Reference for seawater concentration |
|---|---|---|---|---|
| ALK4 | 2 | Yes | 0.12 | Plass-Dülmer et al., 1993 |
| C2H2 | 1 | No | 0.02 | Plass-Dülmer et al., 1993 |
| ISOP | 1 | Yes | 1.64 | Arnold et al., 2009 |
| RCHO | 1 | Yes | 7.47 | Singh et al., 2003 |
| MEK | 1 | Yes | -7.23 | Schlundt et al., 2017 |
| **Total net emission** | | | -68.25 | |
| **Total net emission producing acetaldehyde** | | | 8.23 | |

[1]Net ocean emissions = upward flux out of the ocean - ocean uptake.

[2]More information on the GEOS-Chem species definitions can be found here: http://wiki.seas.harvard.edu/geos-chem/index.php/Species_in_GEOS-Chem.

5    **Table 4.** Abiotic ocean VOC emissions according to Brüggemann et al. (2018)[1]

| GEOS-Chem species[2] | # lumped species | Produces acetaldehyde? | Annual Emission (Tg C) |
|---|---|---|---|
| ACET | 1 | No | 10.07 |
| EOH | 1 | Yes | 5.16 |
| ALD2 | 1 | Yes | 2.26 |
| MOH | 2 | No | 0.79 |
| RCHO | 21 | Yes | 3.88 |
| ISOP | 1 | Yes | 1.04 |
| PRPE | 13 | Yes | 4.44 |
| MACR | 1 | Yes | 0.42 |
| ACTA | 1 | Yes | 0.10 |
| CH2O | 1 | No | 0.03 |
| XYLE | 1 | No | 0.05 |
| TOLU | 1 | No | 0.04 |
| BENZ | 1 | No | 0.02 |
| **Total net emission** | | | 28.30 |
| **Total net emission producing acetaldehyde** | | | 17.29 |





[1]Table S2 shows the emission factor assumed for each species and the lumping methodology for Table 4.
[2]More information on the GEOS-Chem species definitions can be found here: http://wiki.seas.harvard.edu/geos-chem/index.php/Species_in_GEOS-Chem.

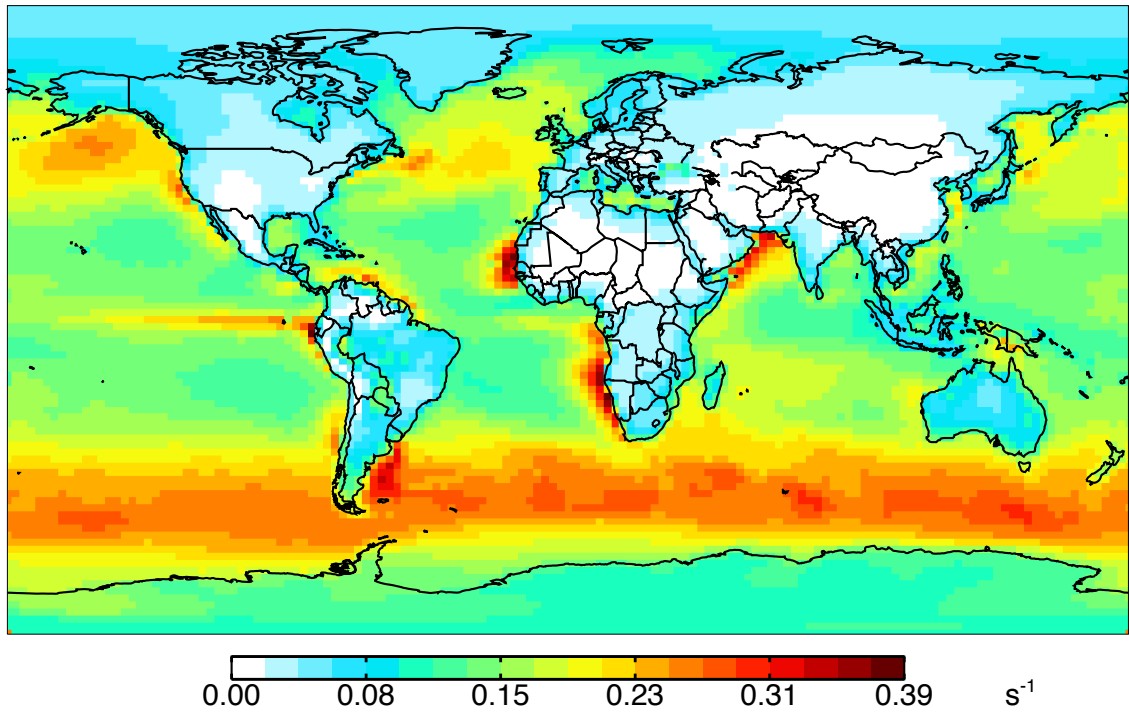

5   **Figure 9.** Impact of all ocean emissions (Tables 3 and 4) on annual simulated 2016 surface cOHR as described in the text.





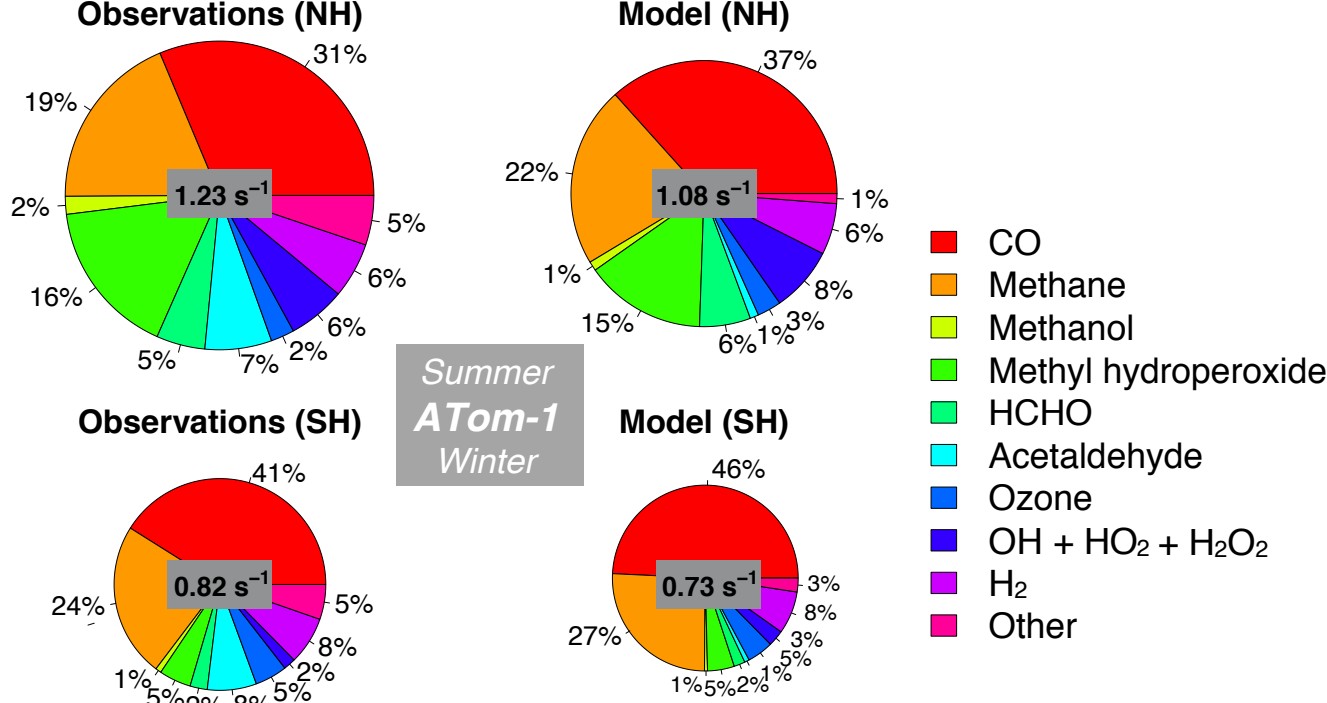

**Figure 10**. Median observed and modeled OHR and cOHR (see text) below 3km in the Northern Hemisphere (>0°N) and Southern Hemisphere (<0°S) during A*T*om-1. The "Other" category the following species as described in Table 2: ethanol, propane, ethane, acetone, >C₃ aldehydes, methyl ethyl ketone, methyl vinyl ketone, methacrolein, benzene, toluene, >C₄ alkanes, peroxyacetic acid, peroxynitric acid, dimethyl sulfide, nitric acid, NO, and NO₂. The diameter of each pie chart is scaled relative to that with maximum cOHR for A*T*om-1.





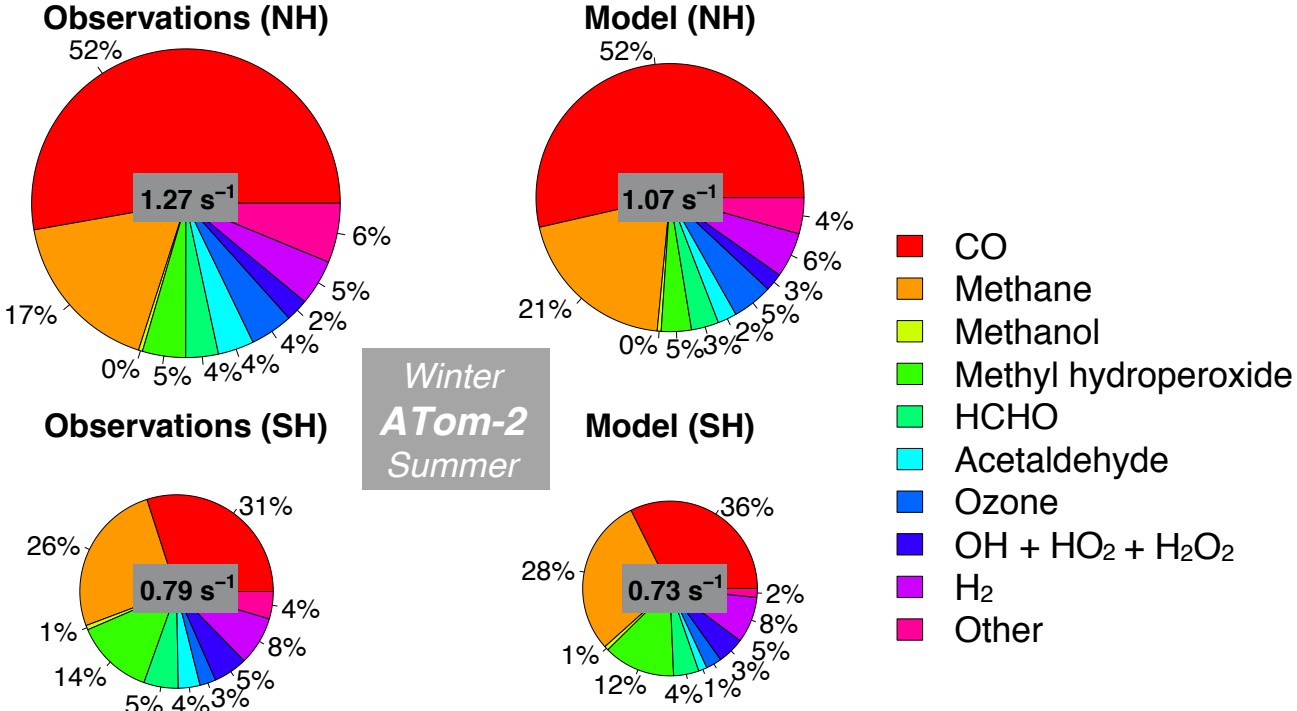

**Figure 11**. Same as Figure 10 but for ATom-2. The diameter of each pie chart is scaled relative to that with maximum cOHR for ATom-2.





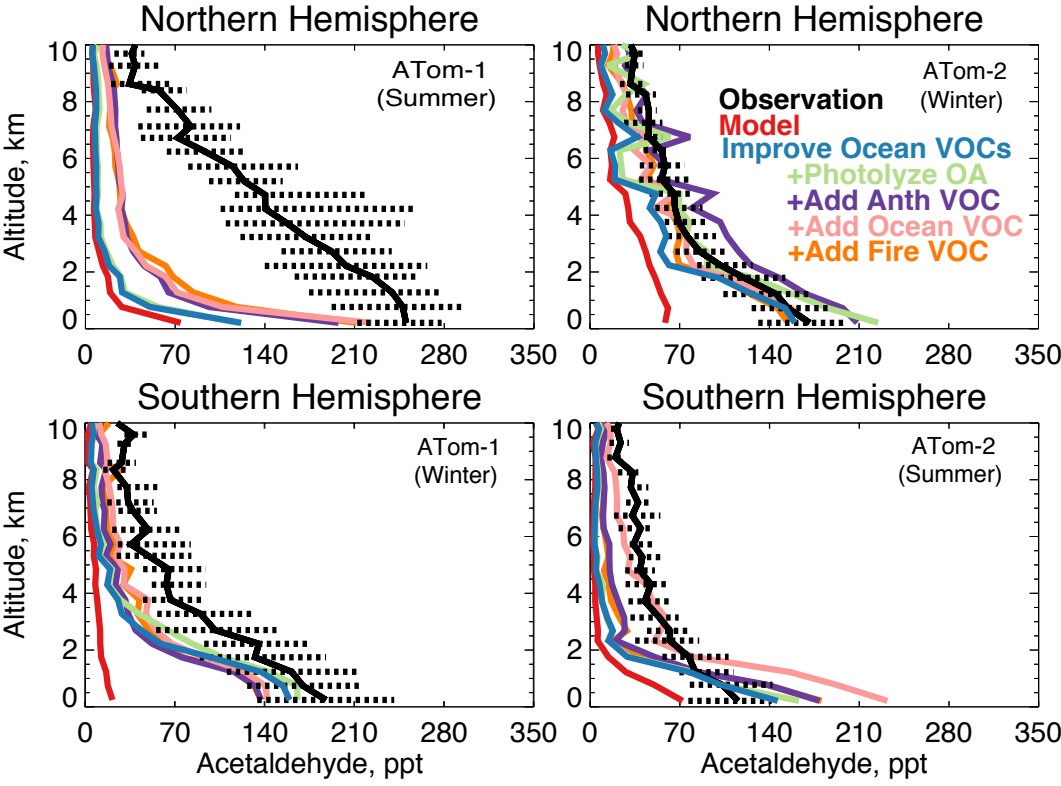

**Figure 12.** The same as Figure 3 for median acetaldehyde profiles. Acetaldehyde was measured by the TOGA instrument as described in Table 2. The sensitivity studies are described in the text.

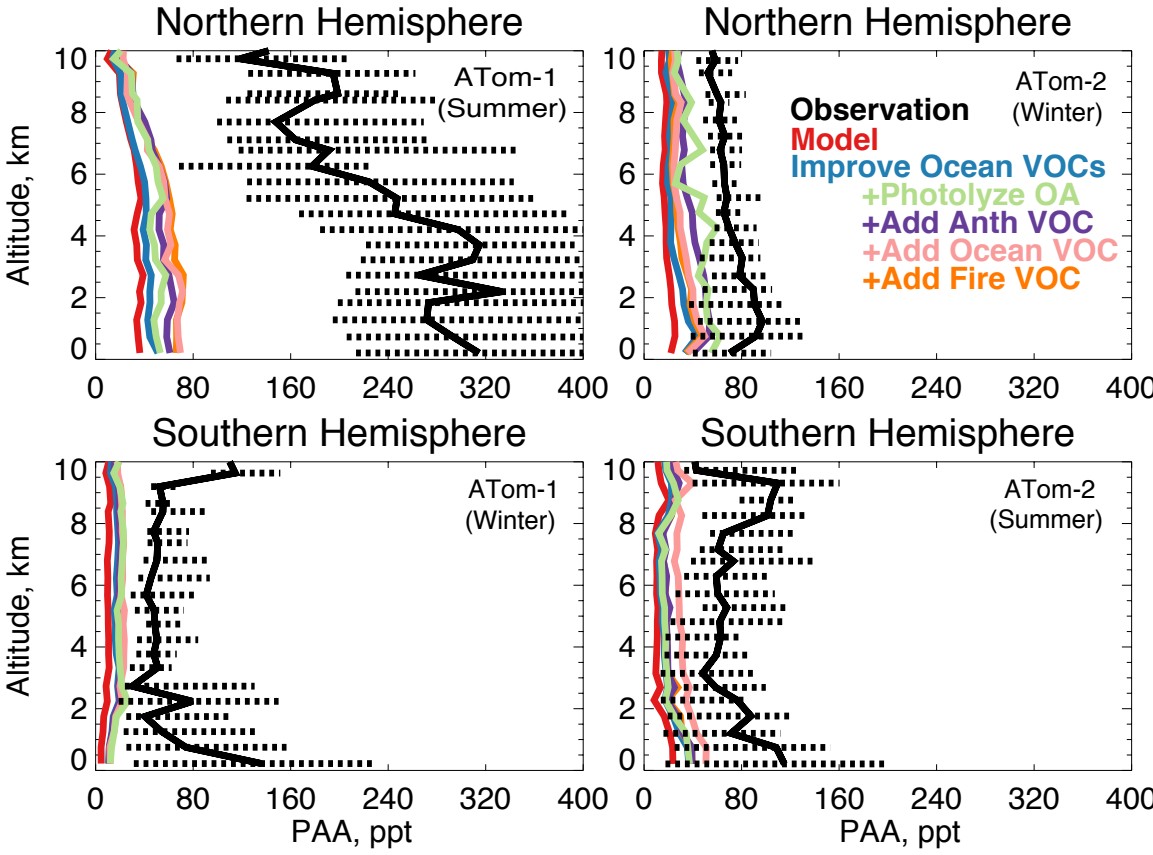

**Figure 13.** The same as Figure 3 for median peroxyacetic acid (PAA) profiles. PAA was measured by the Caltech CIMS instrument as described in Table 2. The sensitivity studies are described in the text.



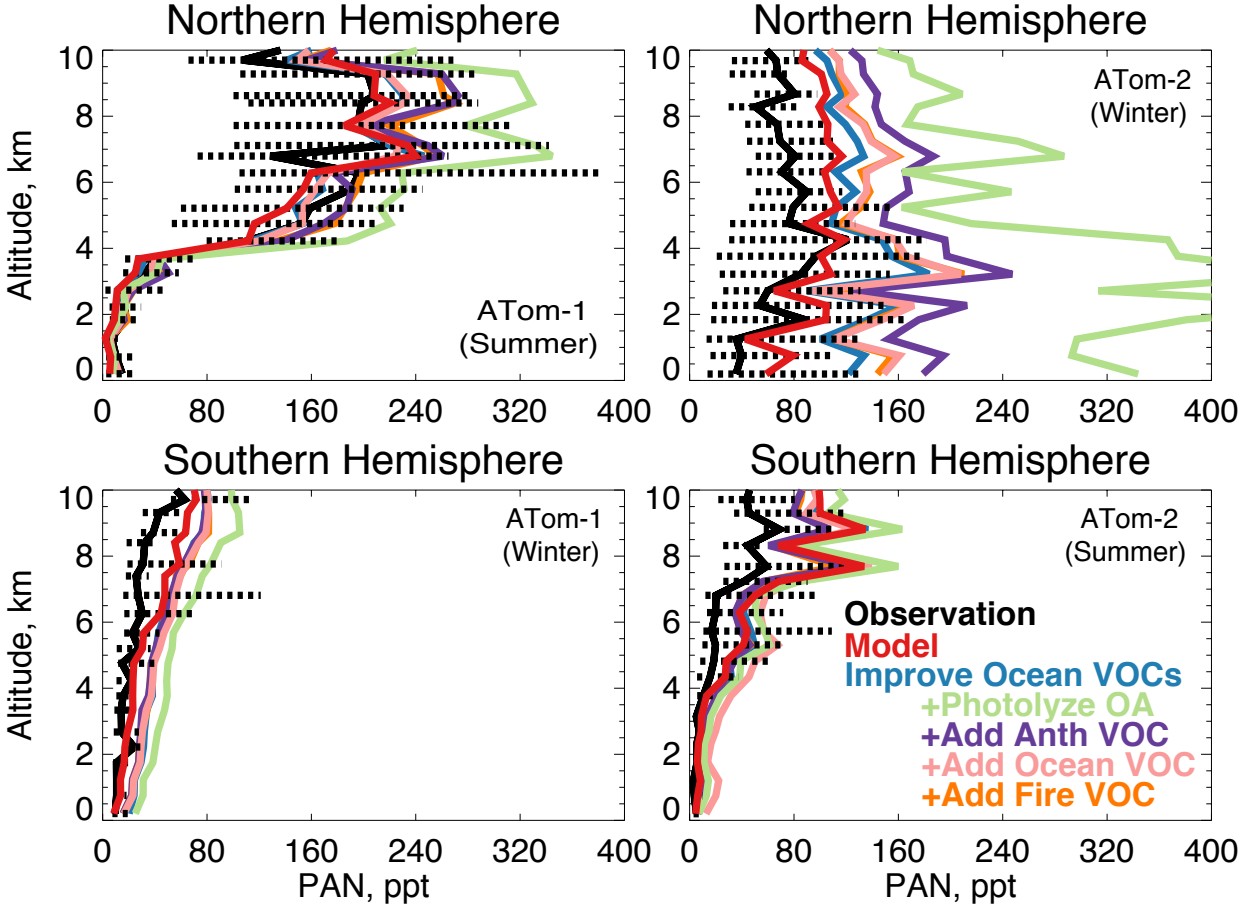

**Figure 14.** The same as Figure 3 for median peroxyacetyl nitrate (PAN) profiles. PAN was measured by the PANTHER instrument as described in Table 2. The sensitivity studies are described in the text.

**Table 5.** Model sources of acetaldehyde in 2016

| Sources (Tg yr$^{-1}$)[1] | Millet et al. (2010) | This Work |
|---|---|---|
| Photochemical production | 128 | 160 |
| Net ocean emission | 57 | 22 |
| Terrestrial plant growth + decay | 23 | 26 |
| Biomass burning | 3 | 3 |
| Anthropogenic emission | 2 | 2 |
| Total source | 213 | 213 |

[1]Emissions are given in Tg of acetaldehyde per year for comparison to Millet et al. (2010). These totals are for the baseline model simulation described in Section 2.1.

