# Peer review of "Constraining remote oxidation capacity with ATom observations"

_Atmospheric Chemistry and Physics, 2019_

## Referee Comment (RC1) · Anonymous Referee #2 · 29 Jan 2020

General Comments

The article "Constraining remote oxidation capacity with ATom observations" by Travis et al., submitted for publication in Atmospheric Chemistry and Physics, presents an evaluation of the hydroxyl radical (OH) and OH reactivity (OHR) measured during the Atmospheric Tomography Mission (ATom) against the GEOS-Chem global chemical transport model. While OH is generally well-modeled in the remote regions sampled by ATom, notable biases occur in wintertime Northern Hemisphere NOy and in OHR in the lowest 3 km. Multiple explanations for these discrepancies are discussed through literature review and explored through model sensitivity simulations.

The submitted manuscript provides a wide-ranging examination of the complex, timely topic of tropospheric oxidizing capacity. While the study does not provide any strong

evidence for solutions to the outstanding questions regarding NOyÂň and OHR biases, it presents a helpful survey of all the issues and demonstrates how a number of them may be further explored in a global modeling framework. Also, the conclusion that GEOS-Chem does "not show systematic bias in the simulation of OH or the drivers of remote OH production" is an important one, suggesting that persistent biases in models' globally integrated methane lifetime must instead be driven by simulated OH abundances over land, as the authors point out. Because the topic is highly relevant, the conclusions are sound but not overstated, and the improvements I would suggest are only moderate, I would suggest publication in ACP once the items noted below are addressed.

Specific Comments

The Introduction does not acknowledge the Thames et al. (ACPD, 2019) manuscript, cited later on Pg. 11, which has a likelihood of being published prior to the finalization of this submission. That paper also discusses the ATom OHR measurements, albeit with a focus on the MBL instead of global oxidation capacity. Since the Thames et al. paper will be so closely related to this one by Travis et al., a discussion of its findings and how Travis et al. will complement Thames et al. is warranted here.

Pg. 2 L. 29: The authors acknowledge "the persistent CO underestimate in models" yet do not go on to evaluate this large sink of OH. A figure analogous to Fig. 3, showing CO comparisons between model and observations should be included and discussed. Does the reasonably accurate OH field within this GEOS-Chem simulation translate to similarly well-simulated fields of CO over the oceans? Or does longer-lived CO have the imprint of biased continental OH, to which the authors refer?

Pg. 4 L. 3: Could you specify whether the methane concentration boundary condition varies with latitude and/or longitude? And, since they derive from monthly observations, is it correct to assume that the boundary condition changes from month to month?

[Figure]

Pg. 4 L. 30: Please describe how exactly the tropospheric mean OH is being calculated. As Lawrence et al., 2001 explain, there are multiple ways to weight this calculation, and, for the purposes of facilitating comparisons of these values between studies, an explicit definition of this metric should be included in each paper that discusses it.

Figures 3-8, 12-14: Please consider trying to visualize not only the 25th/75th percentiles for the observed median profiles, but also for the modeled profiles. How well the spread of each of these quantities agrees can be instructive as well.

On organization: I found, reading through the paper, that the topics of various results/discussion sections (Sections 3-6) jumped around quite a bit. For instance:

-Some discussion of the literature on acetaldehyde is initiated in Section 5 (mention of Read et al., 2012 on Pg. 10, L.4), mentioned again farther down on the page (Pg. 10, L. 29), and continued throughout Section 6. I would suggest consolidating the discussion of the acetaldehyde literature in one place, and perhaps making Section 6 a subsection of Section 5.

-Similarly, the discussion of NOy as a proxy for OH secondary source NOx is understandable, given the issues with measured NO2, but the discussion necessarily turns to HNO3 evaluation, all under Section 4: Constraints on the remote source of OH. Generally, HNO3 is viewed as a sink for OH, so this further contributes to the feeling of "jumping around" between topics.

Additional subsections and improvements in framing the discussion should help to give a more logical structure to these sections .

Technical Corrections

Pg. 2 L. 1: The sentence starting "Comparisons" is a run-on; either include a comma between "aerosol but" or separate into two sentences

Pg. 2 L. 5: Run-on sentence; place comma between "sources and" or split to two sentences.

Pg. 2 L. 20: Run-on sentence; place comma between "atmosphere and" or split.

Pg. 4 L. 1: Place comma after "Sherwen et al."

Pg. 4 L. 25: MCM v3.3.1 has an additional reference, regarding the update from v3.2: Jenkin et al., 2015

Pg. 4 L. 29: Figures should generally be numbered in the order that they appear in the text, even Supplemental figures. Fig. S8 should be moved to S1. Same with Tables (S4 and S5 appear before S1), and Fig. S9 (appears before S5).

Pg. 5 L. 13: Could the authors please state the number of species that are listed in Table S1?

Pg. 6 L. 6: "attitude" should be "altitude"

Pg. 6 L. 11: Should there be units for the accuracy value provided here (molec cm–3)?

Pg. 6 L. 17: Please specify if Fig. S1 shows in situ OH concentrations of column averaged. If it is column averaged, please use the OHcol notation in the text at this location and in the figure.

Pg. 6 L. 22: Instead of "successful" and "success" here, simulation should be described as having "good agreement" or similar wording.

Pg. 7 L. 10 & 15: Replace "successfully"

Pg. 7 L. 30: Run-on sentence; place comma between "2018) and"

Pg. 8 L. 10: Anderson et al. 2014 also indicated a bias in the anthropogenic NOx inventory; please cite that paper here as well.

Pg. 8 L. 18: "higher larger ozone" seems redundant

Pg. 8 L. 21: "free tropospheric" should be "free tropospheric bias"?

Pg. 9 L. 21: "We compare OHR. . ." I would suggest explicitly stating here that "OHR"

refers to directly measured OHR, to avoid confusion.

Pg. 11 L. 4: Thames et al. (2019) does not appear in the reference list.

Pg. 11 L. 17: "…when the lifetime of CO is long." I would consider this circular reasoning; the reason the lifetime of CO is long in the wintertime is because OH concentrations are low.

Pg. 11 L. 24: "OF" should be "of"

Pg. 11 L. 30: Nicely et al. (2016) also recognized the importance of acetaldehyde in explaining model vs. measurement-constrained OH differences, could be cited here.

Pg. 12 L. 23: It would be helpful to state, quantitatively, how large the model bias in PAA is.

Pg. 13 L. 9: It is unclear what the percentage values provided in parentheses refer to–are they percent increases in acetaldehyde from corrections to model ethane/propane, or are they percent yields of acetaldehyde per molecule of ethane/propane oxidized?

Table 2: Please number the superscripts in the order they appear in the table.

Figure 5: Units for jO3 should be $10^{-5}$ s$^{-1}$ instead of 105. Would also be helpful to specify whether this is j(O3 –> O1D + O2) or j(O3 –> O3P + O2).

Fig. 11: I appreciate the difficulty of finding unique color choices for a figure like this, but I find the two shades of green, representing MHP and HCHO, practically indistinguishable on my computer screen (so the problem is likely worse in hard copy). Please adjust one of the two.

References

Anderson, D. C., Loughner, C. P., Diskin, G., Weinheimer, A.,Canty, T., P., Salawitch, R. J., Worden, H. M., Fried, A., Mikoviny, T., Wisthaler, A., and Dickerson, R. R.: Measuredand modeled CO and NOyin DISCOVER-

AQ: An evaluation ofemissions and chemistry over the eastern US, Atmos. Environ.,96, 78–87, doi:10.1016/j.atmosenv.2014.07.004, 2014. https://www.sciencedirect.com/science/article/pii/S1352231014005251?via%3Dihub

Jenkin, M. E., Young, J. C., and Rickard, A. R.: The MCM v3.3.1 degradation scheme for isoprene, Atmos. Chem. Phys., 15, 11433–11459, https://doi.org/10.5194/acp-15-11433-2015, 2015. https://www.atmos-chem-phys.net/15/11433/2015/acp-15-11433-2015.html

Lawrence, M. G., Jöckel, P., and von Kuhlmann, R.: What does the global mean OH concentration tell us?, Atmos. Chem. Phys., 1, 37–49, https://doi.org/10.5194/acp-1-37-2001, 2001. https://www.atmos-chem-phys.net/1/37/2001/

Nicely, J. M., et al. (2016), An observationally constrained evaluation of the oxidative capacity in the tropical western Pacific troposphere, J. Geophys. Res. Atmos., 121, 7461–7488, doi:10.1002/2016JD025067. https://agupubs.onlinelibrary.wiley.com/doi/full/10.1002/2016JD025067

Thames, A. B., Brune, W. H., Miller, D. O., Allen, H. M., Apel, E. C., Blake, D. R., Bui, T. P., Commane, R., Crounse, J. D., Daube, B. C., Diskin, G. S., DiGangi, J. P., Elkins, J. W., Hall, S. R., Hanisco, T. F., Hannun, R. A., Hintsa, E., Hornbrook, R. S., Kim, M. J., McKain, K., Moore, F. L., Nicely, J. M., Peischl, J., Ryerson, T. B., St. Clair, J. M., Sweeney, C., Teng, A., Thompson, C. R., Ullmann, K., Wennberg, P. O., and Wolfe, G. M.: Missing OH Reactivity in the Global Marine Boundary Layer, Atmos. Chem. Phys. Discuss., https://doi.org/10.5194/acp-2019-866, in review, 2019. https://www.atmos-chem-phys-discuss.net/acp-2019-866/

---

## Referee Comment (RC2) · Anonymous Referee #1 · 6 Feb 2020

Review of Atmos. Chem. Phys. Manuscript (#ACP-2019-931) "Constraining remote oxidation capacity with ATom observations" by Travis et al.

**General Comments**

In this work, measurements of OH and OH reactivity (OHR) from two Atom field deployments were used to evaluate the oxidation capacity over the remote oceans and its representation in the GEOS-Chem model. Good model-measurement agreement was obtained for OH and its precursors with over estimation of NOy to be attributed to insufficient or missing loss processes. The measured OHR below 3 km is greater than the sum OHR calculated from the measured OH reactants and the OHR in GEOS-Chem. The underestimate of acetaldehyde and peroxyacetic acid in GEOS-Chem and the reconcile of model-measurement agreement call for further work on the OH loss and production processes over land. In general, the paper is well written and within the scope of ACP. Evaluation of a transport chemical model against observations in a global scale are important and rare. I would recommend accepting it for publication after the authors address the following special comments in their revision.

**Special Comments**

1. P.2, L.1, maybe change "Organic aerosol is …" to "The production of organic aerosols is …" or something like that.

2. P.2, L.21-22: I would suggest adding "In the remote atmosphere" before "OH is primarily produced by the photolysis of ozone ($O_3$) in the presence of water vapor." as in polluted environments other OH sources like the photolysis of HONO could be dominant at certain times of day.

3. In Section 2.1: a brief discussion of the GEOS-Chem model uncertainties in simulating OHR and OH as well other important species like acetaldehyde and PAA and should be included. Coupled with the measurement uncertainties in Table 2, the combined measurement-model uncertainties can help to understand if the discrepancies in the model-measurement comparisons (Figures 3-8 & 10-14) are significant or not.

4. P.5, L.13 and Figure 1(a): define the altitude range of the surface layer.

5. P.5, L.14-15: How are three quarters and 40% calculated? Are they numerical ratios or somehow weighted? In my opinion, spatially integrated and maybe air mass weighted cOHR values (over oceans versus over land and below 3 km versus above 3 km) should be considered in terms of global oxidation capacity.

6. P.5, L.24: Noticed both Fig. and Figure are used. Not sure which one is required by ACP to use, but please be consistent.

7. P.6, L.11: "accuracy of 1.35" is given for OH measurement: what are the units for 1.35? In Table 2, "factor of 1.35" is given for OH and $HO_2$ detection limit and precision but no accuracy is given. Please clarify it.

8. P.6, L.14: the minimal bias of <1% seems too good to be true considering the spatiotemporal variabilities in both model and measurements (see e.g., Figures 3 and S1) as well as uncertainties and discrepancies (e.g., Fig. 6-7) in the measurement and model.

9. P.6, L.15-16: please include standard deviations for these concentrations.

10. Fig. 3-7: not sure if it's going to be too messy, but it is also important to see the percentiles in the model predictions. Maybe use whiskers to show the percentiles with different altitude bins for the measurement and model so that the whiskers will not be overlapped?

11. P.7 L.2: change VOC to VOCs

12. P.7, L.15-16 and Fig. 5: the units for i(O$_3$) should be $10^{-5}$ s$^{-1}$. Also please point out this is for the photolysis reaction of O$_3$ + hv $\rightarrow$ O($^1$D) + O$_2$ (sometimes it's called jO($^1$D), not the other one: O$_3$ + hv $\rightarrow$ O($^3$P) + O$_2$. I pointed this out in the initial review before the ACPD publication, but it seemed the message didn't get through to the authors.

13. P.7, L.17-26: there is a discrepancy as large as 30 ppb at high altitudes for winter in ATom-1 and for winter and summer in Atom-2. A discrepancy of 20-30 ppb doesn't seem "unbiased" to me. Any explanation of this overestimate at high altitudes should be briefly discussed here. This also makes me believe the above <1% bias likely to be coincident or the right answers for the wrong reasons.

14. P.9, L.28-29: note in Table 2 an accuracy of 0.8 s$^{-1}$ and a detection limit/precision of 0.3 s$^{-1}$, which are comparable to the overall differences here. The authors should mention this to remind readers the uncertainty in the measure.

15. P.9 bottom and P.10 top: both r and r$^2$ are used for correlation. Please be consistent. In my opinion, r$^2$ should be used. A scatter plot of the missing OH reactivity against acetaldehyde should be included in the SI to support the strongest relationship.

16. P.10, L.4: …oxygenated VOC**s** (OVOC**s**)

17. P.10, L.11: VOCs. There are many cases where "VOC" should be really "VOCs" and "aerosol" should be really 'aerosols". Please check this out through the manuscript.

18. P.10: Fig. S3 and 9: any explanation why there is a belt of enhancement over the ocean in the mid-latitude of southern hemisphere?

19. P.11, L. 24: change OF to of.

20. Table 2: in the reference for CH$_4$, remove AMT

21. Note 1 of Table 2: H$_2$ was measured but was set to 0.5 ppm. How does this value compare to the observations? If the difference is large, maybe use the measured value (e.g., mean) to constrain the model. From Figures 10 & 11, H$_2$ contributed about 5-8% of observed and modeled OHR, which is not very small.

22. In Supplemental Information: both ethane and propane are underestimated in GEOS-Chem (Figures S6 and S7). Is this because of unaccounted emission sources like fracking?

---

## Author Comment (AC1) · 12 Apr 2020

**Response to Anonymous Referee #1.**

**We thank the reviewer for providing suggestions to improve the manuscript. Our responses to their comments are shown in blue. Added text is shown in italics.**

In this work, measurements of OH and OH reactivity (OHR) from two Atom field deployments were used to evaluate the oxidation capacity over the remote oceans and its representation in the GEOS-Chem model. Good model-measurement agreement was obtained for OH and its precursors with over estimation of NOy to be attributed to insufficient or missing loss processes. The measured OHR below 3 km is greater than the sum OHR calculated from the measured OH reactants and the OHR in GEOS-Chem. The underestimate of acetaldehyde and peroxyacetic acid in GEOS-Chem and the reconcile of model-measurement agreement call for further work on the OH loss and production processes over land. In general, the paper is well written and within the scope of ACP. Evaluation of a transport chemical model against observations in a global scale are important and rare. I would recommend accepting it for publication after the authors address the following special comments in their revision.

Special Comments
P.2, L.1, maybe change "Organic aerosol is …" to "The production of organic aerosols is …" or something like that.
Done.

P.2, L.21-22: I would suggest adding "In the remote atmosphere" before "OH is primarily produced by the photolysis of ozone (O3) in the presence of water vapor." as in polluted environments other OH sources like the photolysis of HONO could be dominant at certain times of day.
We agree with the reviewer that other sources of OH can dominate in polluted environment, however this is not precluded by our statement that "OH is primarily produced by the photolysis of O3" which is an accurate reflection of global OH production. To address the reviewer's comment, we have revised the text on P8, L4 to read *"In the remote troposphere, OH…"* to address this comment.

In Section 2.1: a brief discussion of the GEOS-Chem model uncertainties in simulating OHR and OH as well other important species like acetaldehyde and PAA and should be included. Coupled with the measurement uncertainties in Table 2, the combined measurement-model uncertainties can help to understand if the discrepancies in the model-measurement comparisons (Figures 3-8 & 10-14) are significant or not.
The reviewer raises an interesting point. Quantifying how the uncertainty associated with the myriad processes (i.e. emissions, transport, removal, chemistry for each species…) integrated within a CTM propagates to simulated concentrations (or reactivity) would be a study (or career!) unto itself. We note that the uncertainties on many of these underlying processes are not well characterized themselves (e.g. how uncertain is the wet removal of nitric acid?). Thus, while we agree that in general, model uncertainty quantification could provide insight into model-observation comparisons, it is not straight-forward to estimate these uncertainties. Furthermore, and perhaps more importantly, here we intend to use the model as a tool to evaluate systematic biases that could be the cause of the global mean OH bias across models, as well as the remote nitric acid and acetaldehyde bias across models. Therefore, we do not believe that an assessment of specific model uncertainties is necessary.  We clarify this point in the introduction, P3, line 28. *"We simulate the first two deployments (ATom-1: July-August 2016, ATom-2: January-February 2017) using the GEOS-Chem chemical transport model (CTM) as our tool to explore potential sources of systematic errors that could explain the community-wide model overestimate in global mean OH and underestimate of the methane lifetime."*

4. P.5, L.13 and Figure 1(a): define the altitude range of the surface layer.
We added the following to P4, L8 to address this point: *"The midpoint of the first model layer is 58 m."*

5. P.5, L.14-15: How are three quarters and 40% calculated? Are they numerical ratios or somehow weighted? In my opinion, spatially integrated and maybe air mass weighted cOHR values (over oceans versus over land and below 3 km versus above 3 km) should be considered in terms of global oxidation capacity.
Thank you for this suggestion.  We now calculate cOHR$_{mod}$ below 3 km as an air-mass weighted quantity (P6, L5). *"Approximately 80 % of air-mass weighted cOHR$_{mod}$ resides below 3 km (Fig. 1b)."*

6. P.5, L.24: Noticed both Fig. and Figure are used. Not sure which one is required by ACP to use, but please be consistent.

Figure is only used when beginning a sentence.  Fig. is used otherwise.

7. P.6, L.11: "accuracy of 1.35" is given for OH measurement: what are the units for 1.35? In Table 2, "factor of 1.35" is given for OH and HO2 detection limit and precision but no accuracy is given. Please clarify it.

Changed factor of 1.35 to "*74 % to 135%*" on P6, L29 and in Table 2.

8. P.6, L.14: the minimal bias of <1% seems too good to be true considering the spatiotemporal variabilities in both model and measurements (see e.g., Figures 3 and S1) as well as uncertainties and discrepancies (e.g., Fig. 6-7) in the measurement and model.

We agree that this was confusing. We have removed "minimal bias <1%" and added the following statement on P7, L21: *"As discussed above, model OH is overestimated in the lowest two kilometers during this period but this bias is minimized in the column average."*

9. P.6, L.15-16: please include standard deviations for these concentrations.

These are median values, and thus standard deviations are not appropriate metrics.  We have clarified this in the text on P7, L3: *"We calculate the **median** air-mass weighted column average OH ($OH_{col}$) from Fig. 3 …. "* and L9 *"Median model $OH_{col}$ is within…"*

10. Fig. 3-7: not sure if it's going to be too messy, but it is also important to see the percentiles in the model predictions. Maybe use whiskers to show the percentiles with different altitude bins for the measurement and model so that the whiskers will not be overlapped?

Thank you for the suggestion. We have added the 25th – 75th percentiles to the model as well for Fig. 3-7.

11. P.7 L.2: change VOC to VOCs

Changed.

12. P.7, L.15-16 and Fig. 5: the units for i(O3) should be 10-5 s-1. Also please point out this is for the photolysis reaction of O3 + hv → _O(1D) + O2 (sometimes it's called jO(1D), not the other one: O3 + hv → _O(3P) + O2. I pointed this out in the initial review before the ACPD publication, but it seemed the message didn't get through to the authors.

Changed jO3 to "$jO(^1D)$" in the text and the figure and fixed the units.

13. P.7, L.17-26: there is a discrepancy as large as 30 ppb at high altitudes for winter in ATom-1 and for winter and summer in Atom-2. A discrepancy of 20-30 ppb doesn't seem "unbiased" to me. Any explanation of this overestimate at high altitudes should be briefly discussed here. This also makes me believe the above <1% bias likely to be coincident or the right answers for the wrong reasons.

There is a nice analysis on the contribution of O1D + H₂O to PHOₓ from Brune et al., 2020. We revised this sentence to read P8, L28: *"Upper tropospheric ozone is overestimated in all cases but Northern Hemisphere summer, but this would not have a large influence on primary OH production (or the methane lifetime) at these altitudes (Brune et al., 2020)."*

14. P.9, L.28-29: note in Table 2 an accuracy of 0.8 s-1 and a detection limit/precision of 0.3 s-1, which are comparable to the overall differences here. The authors should mention this to remind readers the uncertainty in the measure.

Thames et al. 2020 (just published in ACP) provide statistical tests of the missing reactivity discussed in this work. We add the following description of their findings on P11, L26. *"Thames et al. (2020) showed that median missing reactivity (between OHR and an observationally-constrained box model) below 4 km during the ATom-1, ATom-2, and ATom-3 deployments was between 0.2 and 0.8 s-1. They provided statistical evidence that while near the level of the instrument accuracy, missing OHR in the marine boundary layer was statistically significant."*

15. P.9 bottom and P.10 top: both r and r2 are used for correlation. Please be consistent. In my opinion, r2 should be used. A scatter plot of the missing OH reactivity against acetaldehyde should be included in the SI to support the strongest relationship.

Only $r^2$ is now used, and a scatter plot of the missing OH reactivity against acetaldehyde is now Figure S3.

16. P.10, L.4: …oxygenated VOCs (OVOCs)

Replaced.

17. P.10, L.11: VOCs. There are many cases where "VOC" should be really "VOCs" and "aerosol" should be really 'aerosols". Please check this out through the manuscript.

Now using "VOCs" and "aerosols".

18. P.10: Fig. S3 and 9: any explanation why there is a belt of enhancement over the ocean in the mid-latitude of southern hemisphere?

This is due to setting a minimum seawater concentration for acetaldehyde. This was perhaps not clear because this adjustment is described later in the manuscript. We add the following clarification to P12, line 22, *"Figure 10 shows the annual mean impact of all ocean emissions described in Tables 3 and 4 (including an adjustment to the acetaldehyde seawater concentration described below in 5.1) on cOHR$_{mod}$…"*

And to P12, line 25, *"The largest increases occur in regions of higher biogenic activity along coastlines and in the Southern Ocean due to the adjustment to acetaldehyde emissions discussed in Section 5.1…"*

19. P.11, L. 24: change OF to of.

Fixed.

20. Table 2: in the reference for CH4, remove AMT

Fixed.

21. Note 1 of Table 2: H2 was measured but was set to 0.5 ppm. How does this value compare to the observations? If the difference is large, maybe use the measured value (e.g., mean) to constrain the model. From Figures 10 & 11, H2 contributed about 5-8% of observed and modeled OHR, which is not very small.

We added the following sentence to the model description on Page 5, L14: *"The model concentration of H$_2$ is fixed at 500 ppt, consistent with observed H$_2$ from ATom-1 and ATom-2 (520 ppt)."*

22. In Supplemental Information: both ethane and propane are underestimated in GEOS-Chem (Figures S6 and S7). Is this because of unaccounted emission sources like fracking?

We added the following citation to P16, L2: *"The model underestimates average ethane and propane below 10 km by 100 % and 40 %, respectively during ATom-2 (Figs. S8 and S9) which could be due to underestimated natural geologic and fossil fuel emissions (Dalsøren et al., 2018).*

**Response to Anonymous Referee #2.**

We thank the reviewer for providing suggestions to improve the manuscript. Our responses to their comments are shown in blue. Added text is shown in italics.

The Introduction does not acknowledge the Thames et al. (ACPD, 2019) manuscript, cited later on Pg. 11, which has a likelihood of being published prior to the finalization of this submission. That paper also discusses the ATom OHR measurements, albeit with a focus on the MBL instead of global oxidation capacity. Since the Thames et al. paper will be so closely related to this one by Travis et al., a discussion of its findings and how Travis et al. will complement Thames et al. is warranted here.

We agree with the reviewer – Thames et al. was very recently published in final peer reviewed form on ACP, and we now comment further on this paper in our own manuscript. Page 3, L14 - We add a citation for Thames et al, 2020 to the Introduction. We add additional discussion of Thames et al, 2020 in the following places:

P3, L19: *"Thames et al. (2020) found evidence of missing OHR between measurements an observationally-constrained box model during the first three ATom deployments."*

P11, L16: *"During ATom, Thames et al. (2020) measured OHR over the Atlantic and Pacific oceans in all four seasons and determined that missing OHR correlated with oxygenated VOCs suggesting the presence of unknown ocean emissions."*

P11, L16: "Thames et al. (2020) showed that median missing reactivity (between OHR and an observationally-constrained box model) below 4 km during the ATom-1, ATom-2, and ATom-3 deployments was between 0.2 and 0.8 s$^{-1}$ and provided statistical evidence that while near the level of the instrument accuracy, missing OHR in the marine boundary layer is statistically significant."

Pg. 2 L. 29: The authors acknowledge "the persistent CO underestimate in models" yet do not go on to evaluate this large sink of OH. A figure analogous to Fig. 3, showing CO comparisons between model and observations should be included and discussed. Does the reasonably accurate OH field within this GEOS-Chem simulation translate to similarly well-simulated fields of CO over the oceans? Or does longer-lived CO have the imprint of biased continental OH, to which the authors refer?
We evaluate model CO in Fig. 11 and Fig. 12. We add the following statement to P13, L29: *"There is no systematic underestimate in CO as might be expected from the general model underestimate of CO described by Shindell et al. (2006) with the exception of a 10 % underestimate during Northern Hemisphere winter when the lifetime of CO is longer and biases in continental sources could have a larger impact."*

Pg. 4 L. 3: Could you specify whether the methane concentration boundary condition varies with latitude and/or longitude? And, since they derive from monthly observations, is it correct to assume that the boundary condition changes from month to month?
We revised the description on P4, L16 to read *"Surface methane concentrations are prescribed monthly using spatially interpolated observations from the NOAA GMD flask network."*

Pg. 4 L. 30: Please describe how exactly the tropospheric mean OH is being calculated. As Lawrence et al., 2001 explain, there are multiple ways to weight this calculation, and, for the purposes of facilitating comparisons of these values between studies, an explicit definition of this metric should be included in each paper that discusses it.
Thank you for this suggestion, we have added a link to the GEOS-Chem calculation of tropospheric mean OH on P5, L16 *"(see http://wiki.seas.harvard.edu/geos-chem/index.php/Mean_OH_concentration for the detailed calculation)."*

Figures 3-8, 12-14: Please consider trying to visualize not only the 25th/75th percentiles for the observed median profiles, but also for the modeled profiles. How well the spread of each of these quantities agrees can be instructive as well.
All figures now include the 25$^{th}$/75$^{th}$ percentiles for the model as well.

On organization: I found, reading through the paper, that the topics of various results/discussion sections (Sections 3-6) jumped around quite a bit. For instance:

-Some discussion of the literature on acetaldehyde is initiated in Section 5 (mention of Read et al., 2012 on Pg. 10, L.4), mentioned again farther down on the page (Pg. 10, L. 29), and continued throughout Section 6. I would suggest consolidating the discussion of the acetaldehyde literature in one place, and perhaps making Section 6 a subsection of Section 5.

We appreciate the reviewer's suggestion to clarify the text. We have changed Section 6 to Section 5.1. We moved all discussion of acetaldehyde in the literature to the first paragraph of Section 5.1.

-Similarly, the discussion of NOy as a proxy for OH secondary source NOx is understandable, given the issues with measured NO2, but the discussion necessarily turns to HNO3 evaluation, all under Section 4: Constraints on the remote source of OH. Generally, HNO3 is viewed as a sink for OH, so this further contributes to the feeling of "jumping around" between topics. Additional subsections and improvements in framing the discussion should help to give a more logical structure to these sections .

We added a subsection for the discussion of HNO$_3$ evaluation: *"4.1 Causes of the remote model bias in HNO3"*

Technical Corrections

Pg. 2 L. 1: The sentence starting "Comparisons" is a run-on; either include a comma between "aerosol but" or separate into two sentences

We separated into two sentences.

Pg. 2 L. 5: Run-on sentence; place comma between "sources and" or split to two Sentences

Split into two sentences.

Pg. 2 L. 20: Run-on sentence; place comma between "atmosphere and" or split.

Placed a comma.

Pg. 4 L. 1: Place comma after "Sherwen et al."

Placed a comma.

Pg. 4 L. 25: MCM v3.3.1 has an additional reference, regarding the update from v3.2: Jenkin et al., 2015

Added.

Pg. 4 L. 29: Figures should generally be numbered in the order that they appear in the text, even Supplemental figures. Fig. S8 should be moved to S1. Same with Tables (S4 and S5 appear before S1), and Fig. S9 (appears before S5).

The supplement has been re-ordered to follow the order in the text.

Pg. 5 L. 13: Could the authors please state the number of species that are listed in Table S1?

We added the following to P6, L4 "*ninety simulated constituents*…"

Pg. 6 L. 6: "attitude" should be "altitude"

Changed.

Pg. 6 L. 11: Should there be units for the accuracy value provided here (molec cm–3)?

Clarified value as 74% to 135%, 2σ confidence level.

Pg. 6 L. 17: Please specify if Fig. S1 shows in situ OH concentrations of column averaged. If it is column averaged, please use the OHcol notation in the text at this location and in the figure.

Changed "OH" to *"OH concentrations".*

Pg. 6 L. 22: Instead of "successful" and "success" here, simulation should be described as having "good agreement" or similar wording.

We removed this sentence and rephrased our statement on P6, L30. "*This result from a global CTM is consistent with good agreement between OH measurements and a box model during NASA's Pacific Exploratory Mission-Tropics (PEM-Tropic B) campaign in the clean remote Pacific (Tan et al., 2001) and a similar analysis by Brune et al. (2020) for ATom 1 through 4.*"

Pg. 7 L. 10 & 15: Replace "successfully"
Replaced "successfully" with *"reproduces"*

Pg. 7 L. 30: Run-on sentence; place comma between "2018) and"
Added this comma.

Pg. 8 L. 10: Anderson et al. 2014 also indicated a bias in the anthropogenic NOx inventory; please cite that paper here as well.
Added this citation.

Pg. 8 L. 18: "higher larger ozone" seems redundant
We have removed this text and incorporated the simulation of Wang et al., 2019 into our simulation.

Pg. 8 L. 21: "free tropospheric" should be "free tropospheric bias"?
Changed "free tropospheric" to *"free tropospheric bias"*.

Pg. 9 L. 21: "We compare OHR: : :" I would suggest explicitly stating here that "OHR" refers to directly measured OHR, to avoid confusion.
We changed "We compare OHR" to *"We compare directly measured OHR"*.

Pg. 11 L. 4: Thames et al. (2019) does not appear in the reference list.
Thames et al. (2020) has been added to the reference list.

Pg. 11 L. 17: ": : :when the lifetime of CO is long." I would consider this circular reasoning; the reason the lifetime of CO is long in the wintertime is because OH concentrations are low.
We have revised the text to read on P13, L28: *"CO and methane make up half or greater of both cOHRobs and cOHRmod"*.

Pg. 11 L. 24: "OF" should be "of"
Fixed.

Pg. 11 L. 30: Nicely et al. (2016) also recognized the importance of acetaldehyde in explaining model vs. measurement-constrained OH differences, could be cited here.
We added this citation.

Pg. 12 L. 23: It would be helpful to state, quantitatively, how large the model bias in PAA is.
We changed the sentence on P15, L17 to read "*Figure 14 shows the average model underestimate of below 3 km of 70 to 90 % (60 to 250 ppt)."*

Pg. 13 L. 9: It is unclear what the percentage values provided in parentheses refer to– are they percent increases in acetaldehyde from corrections to model ethane/propane, or are they percent yields of acetaldehyde per molecule of ethane/propane oxidized?
Changed "currently…" to *"model yields are.."*

Table 2: Please number the superscripts in the order they appear in the table.
The subscripts have been re-ordered.

Figure 5: Units for jO3 should be 10–5 s–1 instead of 105. Would also be helpful to specify whether this is j(O3 –> O1D + O2) or j(O3 –> O3P + O2).
Fixed the units and changed $jO_3$ to $jO^1D$.

Fig. 11: I appreciate the difficulty of finding unique color choices for a figure like this, but I find the two shades of green, representing MHP and HCHO, practically indistinguishable on my computer screen (so the problem is likely worse in hard copy). Please adjust one of the two.

The color of HCHO has been adjusted to a darker green.

[revised manuscript text omitted]